# Spatially distinct epithelial and mesenchymal cell subsets along progressive lineage restriction in the branching embryonic mammary gland

Claudia Carabaña[1,2,4], Wenjie Sun [ID][1,5], Camila Veludo Ramos[1,5], Mathilde Huyghe [ID][1],
Meghan Perkins [ID][1], Aurélien Maillot [ID][1], Robin Journot[1], Fatima Hartani[1], Marisa M Faraldo [ID][1],
Bethan Lloyd-Lewis [ID][3,6 ✉] & Silvia Fre [ID][1,6 ✉]

## Abstract

**How cells coordinate morphogenetic cues and fate specification during development remains a fundamental question in organogenesis. The mammary gland arises from multipotent stem cells (MaSCs), which are progressively replaced by unipotent progenitors by birth. However, the lack of specific markers for early fate specification has prevented the delineation of the features and spatial localization of MaSC-derived lineage-committed progenitors. Here, using single-cell RNA sequencing from E13.5 to birth, we produced an atlas of matched mouse mammary epithelium and mesenchyme and reconstructed the differentiation trajectories of MaSCs toward basal and luminal fate. We show that murine MaSCs exhibit lineage commitment just prior to the first sprouting events of mammary branching morphogenesis at E15.5. We identify early molecular markers for committed and multipotent MaSCs and define their spatial distribution within the developing tissue. Furthermore, we show that the mammary embryonic mesenchyme is composed of two spatially restricted cell populations, and that dermal mesenchyme-produced FGF10 is essential for embryonic mammary branching morphogenesis. Altogether, our data elucidate the spatiotemporal signals underlying lineage specification of multipotent MaSCs, and uncover the signals from mesenchymal cells that guide mammary branching morphogenesis.**

**Keywords** Mammary Stem Cells; Fate Specification; scRNA-seq;
Branching Morphogenesis; Embryonic Explants
**Subject Categories** Development; Methods & Resources; Stem Cells &
Regenerative Medicine

## Introduction

To generate functional organs, cell fate acquisition, and multicellular morphogenetic events must be tightly coordinated. Accordingly, lineage commitment encompasses a progressive differentiation process dictated by transcriptional and mechanical changes to drive the formation of specialist tissues of complex shapes and functions (Chan et al, 2017). The development of the branched mammary gland (MG) is a case in point, being initially formed from multipotent embryonic mammary stem cells (MaSCs) which reorganize through individual and collective movements during branching morphogenesis until committing to specific luminal and basal lineages at birth. Subsequently, unipotent progenitors drive adult homeostasis (van Amerongen et al, 2012; Blaas et al, 2016; Davis et al, 2016; van Keymeulen et al, 2011; Lilja et al, 2018; Lloyd-Lewis et al, 2018; Prater et al, 2014; Scheele et al, 2017; Wuidart et al, 2016, 2018). The embryonic mammary gland, therefore, represents a powerful tissue paradigm to study the integration of stem cell fate specification with tissue morphogenesis.

Mouse mammary gland development begins at embryonic day (E) 10 with the formation of bilateral milk lines, followed by the asynchronous appearance of five pairs of epithelial placodes positioned symmetrically at each side of the embryo (Mailleux et al, 2002; Veltmaat et al, 2004). By E13, these placodes invaginate into the underlying mesenchyme to give rise to mammary buds. At around E15.5, the epithelium undergoes the first sprouting events to invade the underlying fat pad precursor, triggering branching morphogenesis and the formation of a small rudimentary ductal tree by birth (reviewed in (Watson and Khaled, 2020)). The adult mammary ductal network is composed of a bi-layered epithelium comprising two main cell types: an outer layer of myoepithelial or basal cells (BCs), in contact with the basement membrane (BM), and an inner layer of polarized cuboidal luminal cells (LCs), facing the ductal lumen. At puberty, luminal cells can be further subdivided in two populations, depending on their expression of the hormone receptors Estrogen-α (ERα) and Progesterone (PR),

[1]Institut Curie, Laboratory of Genetics and Developmental Biology, PSL Research University, INSERM U934, CNRS UMR3215, 75248 Paris, France. [2]Sorbonne Université, Collège Doctoral, Paris, France. [3]School of Cellular and Molecular Medicine, University of Bristol, Biomedical Sciences Building, Bristol BS8 1TD, UK. [4]Present address: Department of Health Sciences, Faculty of Biomedical and Health Sciences, Universidad Europea de Madrid, C/Tajo, s/n, Villaviciosa de Odón, 28670 Madrid, Spain. [5]These authors contributed equally to this work as second authors: Wenjie Sun, Camila Veludo Ramos. [6]These authors jointly supervised and contributed equally to this work as senior authors: Bethan Lloyd-Lewis, Silvia Fre. ✉E-mail: bethan.lloyd-lewis@bristol.ac.uk; silvia.fre@curie.fr

termed mature luminal (ML) cells (ERα+/PR+) and luminal progenitor (LP) cells (ERα−/PR−).

Previous lineage-tracing studies using neutral or gene-specific promoters suggested that embryonic/fetal MaSCs may become biased toward a luminal or basal fate prior to birth (van Amerongen et al, 2012; Blaas et al, 2016; Elias et al, 2017; Lilja et al, 2018; Lloyd-Lewis et al, 2018; Wuidart et al, 2018). Our recent work suggested that this lineage bias occurs progressively within a narrow developmental window around embryonic day E15.5, a surprisingly early time in mammogenesis (Lilja et al, 2018). Strikingly, this bias towards luminal and basal cell fates coincides with the remarkable epithelial remodeling that occurs during the first embryonic mammary branching event. Yet, the precise timing of lineage specification during this crucial stage of mammary gland morphogenesis remains unclear, hampered by the lack of specific markers for early fate specification.

It is now well-established that cell fate-specific changes in gene expression can modify the properties of growing tissue and affect its morphogenesis and patterning. In the mammary epithelium, recent studies performed single-cell RNA sequencing (scRNA-seq) analysis at distinct stages of mammary embryonic development and proposed a model whereby multipotent MaSCs drive the earliest stages of mammogenesis. These studies identified subsets of embryonic mammary cells characterized by "hybrid" transcriptional signatures and harboring concomitant expression of luminal and basal genes (Giraddi et al, 2018; Wuidart et al, 2018). In contrast, alternative scRNA-seq studies suggested that only Mammary Epithelial Cells (MECs) with basal characteristics are present in the embryonic gland, and that these bipotent progenitors generate mammary luminal cells postnatally (Pal et al, 2021). Recent single nucleus Assay for Transposase Accessible Chromatin sequencing (snATAC-seq) analyses, however, revealed that MECs at E18.5 exhibit either a basal-like or luminal-like chromatin accessibility profile, suggesting the potential priming of these cells to a lineage-restricted state prior to birth (Chung et al, 2019).

Given these uncertainties, we sought to further define the potency of mammary stem cells and the timing of fate acquisition with spatiotemporal resolution during embryonic mammary morphogenesis, by coupling in vivo single-cell transcriptional mapping at different developmental timescales with ex vivo live imaging of mammary embryonic cell dynamics and fate acquisition during branching morphogenesis. This enabled us to finely dissect the heterogeneity of the mammary gland epithelium throughout embryonic development and define the transcriptional programs orchestrating the progressive lineage restriction of multipotent MaSCs to unipotent progenitors. Importantly, our integrative approach prospectively identified new early markers for commitment toward specific mammary lineages and provided fundamental insights into the position and molecular signatures of resident mammary embryonic mesenchymal cells that communicate with epithelial cells to direct branching morphogenesis and influence cell fate decisions.

## Results

### Lineage restriction is a progressive developmental process

How changes in mammary tissue architecture translate into differential gene expression patterns that are characteristic of specific lineages during development remains unknown in many tissue contexts. To address this in the MG, we performed scRNA-seq analysis of mouse embryonic mammary tissues at four developmental times spanning mammary bud invagination (E13.5), initial sprouting events at the presumptive onset of lineage segregation (E14.5 and E15.5) (Lilja et al, 2018) and postnatal branching morphogenesis (at birth or postnatal day 0, P0) (Fig. 1A). At each timepoint, we microdissected mammary buds from female mouse embryos (pooling tissues from 7 to 12 embryos isolated from different pregnant dams) and isolated mammary epithelial (EpCAM+) and stromal (EpCAM−) cells by FACS for scRNA-seq using the 10x Chromium platform. Basal and luminal subpopulations are indistinguishable in embryonic mammary glands using the EpCAM and CD49f gating strategies routinely applied to adult tissues (Fig. EV1A).

Using the Seurat R package (Stuart et al, 2019), unsupervised clustering of single-cell expression data revealed distinct cell clusters at E13.5, E14.5, E15.5, and P0, respectively (Fig. EV1B), which were manually annotated by matching enriched gene sets with known markers of mammary epithelium, mesenchyme and skin cells. With the objective of mapping MECs undergoing lineage commitment early in embryogenesis, we removed contaminating skin cells (Fig. EV1B) and performed a sub-clustering analysis of epithelial populations (MECs) at each developmental time. A cluster composed of proliferative epithelial cells was identified at E15.5, based on a list of cell cycle-related genes. Variation intrinsic to cell cycle stage was reduced by linearly regressing the annotated S and G2/M scores in further analyses (Fig. EV1C,D). While this analysis identified a single population of MECs at the early E13.5 and E14.5 developmental times, 3 transcriptionally distinct cell clusters were apparent at E15.5 and P0 (Figs. 1B–D and EV1B). Proliferating cells were uniformly distributed among the three E15.5 MECs clusters. The detection of three MECs clusters at E15.5 was intriguing, as our previous studies indicated a commitment toward unipotent cell fate around this developmental stage (Lilja et al, 2018). To investigate this further, we calculated a single-cell ID score for "basal-like" and "luminal-like" cells based on published transcriptomic datasets of adult MECs (Kendrick et al, 2008). A higher single-cell ID score reflects increasing similarity to the reference cell type: adult basal or luminal cells. Interestingly, this analysis revealed that E15.5 MECs could already be resolved into three distinct groups: luminal-like cells, basal-like cells, and a hybrid cell population co-expressing luminal and basal genes (Figs. 1B,C and EV1E). As expected, lineage markers commonly used to distinguish LCs (Krt8, Krt18) from BCs (Krt5, Trp63) in the postnatal mammary gland were co-expressed in all three MECs clusters at E15.5 (Fig. EV1F). Other genes with reported lineage-specific expression (Kendrick et al, 2008) but no well-established role in adult LCs (Anxa1, Ly6d) and BCs (Lmo1, Pthlh, Cxcl14) were found to be already differentially expressed at E15.5, identifying them as potential early lineage markers. Importantly, the basal-like cluster, but not luminal-like cells, was particularly enriched in genes encoding for factors known to be essential for mammary embryonic morphogenesis (Pthlh, Wnt10b, Sostdc1, Bmp2, Msx1). Furthermore, novel genes with no reported expression in MECs were also found specifically in the basal-like cluster (Ptp4a1, Fam60a, Ralbp1) or in the luminal-like cluster (Grhl3, Krtdap).

By applying a computed ID score for each epithelial adult cell type (Kendrick et al, 2008) to the 3 transcriptionally distinct cell populations observed at P0 (Fig. 1D), BCs (Acta2+, Myh11+),

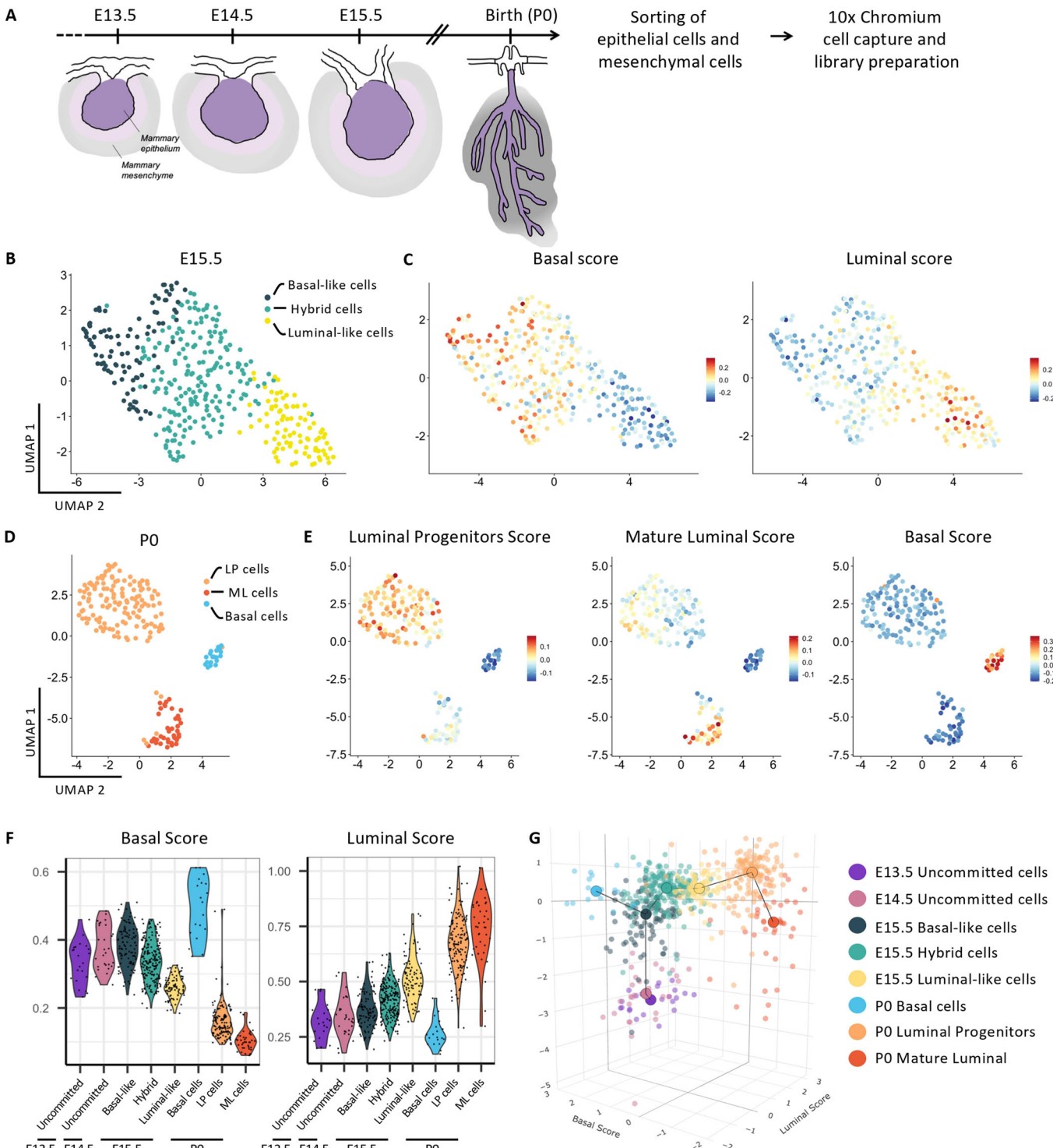

luminal progenitors (LP) (*Notch1+, Aldh1a3+, Lypd3+*) and mature luminal (ML) cells (*Prlr+, Cited1+, Esr1+*) could be clearly distinguished (Fig. 1E). This corroborates our previous findings indicating that MECs are already committed to three distinct lineages at birth (Lilja et al, 2018) and is consistent with previous snATAC-seq analyses of the embryonic mammary gland, which also identified three separate clusters at E18.5 (Chung et al, 2019).

Collectively, our data support a model whereby mammary epithelial cell lineages are progressively being specified throughout development and are well segregated at birth.

We next ordered the cells along pseudo-temporal trajectories to infer the differentiation path of embryonic MECs towards luminal or basal fate. Since we observed that the 2nd principal component of the PCA was highly correlated to the age of the embryos analyzed,

**Figure 1.  Developmental atlas of the transcriptional signatures and 3D trajectory analysis of luminal and basal differentiation of single mammary epithelial cells from E13.5 until birth.**

(A) Scheme showing the isolation and sequencing strategy of mammary embryonic cells at four developmental stages spanning embryonic MG development. (B) UMAP plot of embryonic MECs isolated at E15.5 after subset analysis of non-proliferative MG epithelial cells. Cells are color-coded by cluster. (C) UMAP plots from (B) color-coded according to the expression of the single-cell ID scores in MECs: basal score (left) and luminal score (right). (D) UMAP plot of MECs isolated at P0 after subset analysis of MG epithelial cells. (E) UMAP plots from (D) color-coded according to the expression of luminal progenitors (LP), mature luminal (ML) and basal cell (BC) scores. (F) Violin plots showing the expression levels of the basal and luminal scores in each cluster. $n = 22$ cells at E13.5; $n = 28$ cells at E14.5; $n = 98$ basal-like cells, $n = 199$ hybrid cells and $n = 86$ luminal-like cells at E15.5; $n = 19$ basal cells, $n = 140$ LP cells and $n = 39$ ML cells at P0. Statistical significance was assessed with Wilcoxon test. (G) 3D trajectory of MECs from E13.5 at the origin of the mammary cellular hierarchy to P0 MECs positioned at the end of two divergent differentiation routes. Basal and luminal trajectories were calculated using a minimum spanning tree (MST) connecting the center (in the 3D space) of each cell cluster.

we used it as a proxy for the developmental stage ($y$ axis) and plotted it against the basal and luminal scores computed above (Fig. 1C) on the $x$ axis (Kendrick et al, 2008) (Figs. 1F,G and EV1G). The resulting plot indicates, as predicted, that E13.5 mammary cells lie at the origin of the mammary cellular hierarchy, with E15.5 cell populations occupying intermediate positions and P0 MECs positioned at the end of two divergent trajectories, representing the binary cell fate choice between basal or luminal differentiation. Remarkably, we noticed that basal-like cells at E15.5 can either transition towards the P0 basal cluster, or to a hybrid cell state that will give rise to LCs (Fig. 1G), suggesting that basal-like cells might lie at the origin of both lineages. Alongside, we generated an original web tool of the scRNA-seq dataset that facilitates the interactive visualization of gene expression profiles along the mammary differentiation trajectory (see "Methods", "Data availability", and "Software availability").

Together, our temporal scRNA-seq atlas reveals the molecular changes associated with progressive lineage restriction and identifies subsets of MECs that are already biased towards basal or luminal cell fate at embryonic day E15.5. Thus, both committed (i.e., conceivably unipotent) and undifferentiated (putative multi-potent) cells likely exist at this important developmental stage in mammogenesis, which coincides with the first morphogenetic events of mammary epithelial branching and duct elongation (Lilja et al, 2018).

## Luminal and basal progenitors are already spatially segregated at E15.5

We next sought to identify differentially expressed genes for each mammary epithelial cluster by examining their dynamic expression profile towards luminal or basal differentiation trajectories. While our compiled scRNA-seq atlas emphasized the cellular hetero-geneity of the embryonic mammary epithelium, this extended analysis identified different patterns of expression along the process of basal (Fig. EV2A) or luminal (Fig. EV2B) differentiation throughout embryonic development (from E13.5 to P0).

On the basal trajectory, we found five distinct patterns of expression. Patterns 1 and 2 contained genes with sustained increased expression in early embryonic developmental times, at E13.5 and E14.5. Known key regulators of mammary bud epithelial cells are highly expressed only during early embryonic development, including *Ndnf, Pthlh, Msx1, Tbx3, Sostdc1*, whose expression is lost before birth. Moreover, multiple Wnt-related genes, such as *Wnt3, Wnt6*, and *Fzd10*, were enriched at these early developmental stages.

A different subset of genes, mostly related to cell migration (*Ptp4a1, Fam60a, Ralbp1*), appeared to be transiently upregulated at E15.5 (Pattern 3). Transcripts involved in mammary basal differentiation were progressively increasing towards the P0 basal cluster (Pattern 4); these included myosin-related proteins (*Myl6, Myl9, Myh11, Mylk*) and genes associated with extracellular matrix (ECM) composition and organization (*Lama4, Adamts4, Itga1, Col9a1, Col4a1, Col11a1, Col16a1*). In addition, towards the P0 basal cluster, we also found increased expression levels of genes regulating cell proliferation (*Top1, Cdkn1a, Runx1, Fosl1*), cytoskeletal organization (*Tuba1c, Tubb6*) and angiogenesis (*Tnfrsf12a, Serpine1, Tgfa, Hbegf*) in Pattern 5, suggesting that epithelial growth is highly regulated at this developmental stage.

On the other hand, we observed seven distinct expression patterns along the luminal differentiation trajectory. As expected, the pattern exhibiting increasing expression across the luminal developmental trajectory contains genes with known luminal characteristics, such as *Krt8, Krt18*, and *Krt19* (Pattern 1). A second group of genes that is switched on during late stages of differentiation is enriched for ML cells markers, such as *Cited1* and *Prlr* (Pattern 2). Genes expressed at the beginning of the differentiation process and subsequently repressed along the luminal trajectory include typical basal markers, such as *Krt5* and *Krt14* (Pattern 4). *Sox11* also presents this dynamic pattern of expression, gradually decreasing along the differentiation process. Indeed, *Sox11* is expressed in MECs only during the early stages of MG embryonic development—when MG epithelial cells are largely quiescent—and is no longer detected by E16.5, consistent with our results. Of interest, *Sox11* has been recently involved in cell fate regulation in the embryonic MG (Tsang et al, 2021). Genes involved in epithelial stratification, such as *Lgals7, Dsc3*, and *Krtdap*, are switched on only in luminal-like cells at E15.5 (Pattern 6). Finally, Pattern 7 comprises genes encoding for several Heat shock proteins (Hsps). There is growing evidence that Hsps may impact neurodevelopment through specific pathways regulating cell differentiation, migration or angiogenesis (Miller and Fort, 2018).

To further extend our analysis, we next sought to computation-ally predict specific paracrine interactions between the luminal-like and basal-like cells identified at E15.5 and between more committed LCs (LP and ML) and BCs at P0, using CellPhoneDB, a bioinformatic tool designed to predict significant ligand–receptor interactions between two cell types from scRNA-seq data (Vento-Tormo et al, 2018). This additional analysis indicates that one of the major pathways governing the crosstalk among different MECs is Notch signaling, with Notch receptors expressed in luminal-like cells and Notch ligands (i.e., Jagged2) being enriched in basal-like

cells already at E15.5 (Fig. EV2C). This finding is in agreement with work from our lab and others that implicates Notch signaling as an essential pathway for dictating the binary decisions between luminal and basal cell fates (Lilja et al, 2018). The CellPhoneDB analysis also highlighted the communication between different Eph receptors expressed by basal-like cells (i.e., *Epha7*) and several Ephrins (such as *Efna1*) enriched in luminal-like cells at E15.5. This is interesting since Ephrin signaling plays an important role in cell guidance during embryonic development (White and Getsios, 2014). Finally, our results show that numerous components of the Wnt pathway are expressed in the E15.5 mammary epithelium. Corroborating these findings, our analysis identified *Wnt3* exclusively expressed in BC-like cells, whereas *Wnt4* is enriched in LC-like cells (Fig. EV2C). Interestingly, at P0, some of these inter-epithelial interactions are predicted to be strongly attenuated (Wnt, Ephrin/Eph pathways), while other signals known to operate in the adult gland clearly emerge (EGFR/AREG, Kit, TNF) (Centonze et al, 2020; McBryan et al, 2008; Regan et al, 2012).

Next, we investigated whether lineage bias is reflected by spatial segregation of cells acquiring luminal or basal characteristics during embryonic development. To select transcripts potentially representing novel markers of early fate commitment, we sought to analyze genes that were differentially expressed in specific epithelial clusters at E15.5 (basal-like, luminal-like, and hybrid cells) (Fig. EV3A,B), and/or exhibited a lineage-specific expression pattern along the differentiation trajectories (Figs. EV2A,B and 2A–C). Using these criteria, we identified *Cxcl14, Pthlh*, and *Ndnf* for basal and *Lgals3, Anxa1*, and *Plet1* for luminal lineage specification (Figs. 2A,B and EV3A,B). We subsequently examined the spatiotemporal expression pattern of these selected genes at distinct stages of mammary embryonic development using single-molecule RNA-fluorescence in situ hybridization (smRNA-FISH). Probes for the luminal-specific membrane-associated protein Annexin A1 (*Anxa1*) (Fankhaenel et al, 2023) and the basal-specific secreted chemokine *Cxcl14* (Sjöberg et al, 2016) revealed that at early embryonic stages (E13.5), *Cxcl14* is expressed in all MECs, and *Anxa1* is lowly expressed in rare cells homogeneously distributed within the mammary bud (Fig. 2D–F). However, at the critical developmental time of E15.5, the transcripts for these two genes show divergent spatial distribution patterns, with *Anxa1* expression being mainly confined to cells in the inner bud region and *Cxcl14* transcripts restricted to the external cell layers in contact or close proximity with the BM (Fig. 2D–F). At birth (P0), as expected, *Anxa1* and *Cxcl14* showed clear luminal and basal restricted expression, respectively (Fig. EV3C). To accurately measure the spatial distribution of expression of each marker within single cells, we segmented individual mammary epithelial cells and quantified the number of *Anxa1* and *Cxcl14* transcripts (represented by each dot) per cell (see "Image analysis and quantification" for details). We subsequently computed the ratio of both RNA probes and color-coded each cell based on the proportion of *Anxa1* and *Cxcl14* gene expression (Fig. 2E). To assess statistical significance of our analysis, we then divided each image of mammary bud into three regions of interest (ROIs) represented by concentric rings (outer, middle and internal regions) (Fig. EV3D) and counted the number of RNA molecules within each ring for both markers. This unbiased approach confirmed the uniform expression pattern of *Anxa1* and *Cxcl14* transcripts in all three regions of the mammary bud at E13.5

(Fig. 2F). By E15.5, however, *Anxa1* transcripts were prominently restricted to the middle and inner ring, while *Cxcl14* transcripts appeared preferentially localized to the middle and outer ring of the mammary bud (Fig. 2F). Analogous smRNA-FISH analysis of E15.5 mammary buds with additional probes indicated that *Ndnf* and *Pthlh* are also expressed in embryonic basal-committed MECs, while *Plet1* and *Lgals3* expression likely mark cells biased toward the luminal lineage (Fig. EV3E-F), further corroborating our temporal scRNA-seq analysis (Fig. EV2A,B). Thus, *Anxa1, Plet1, Lgals3* and *Cxcl14, Ndnf, Pthlh* represent novel markers of MECs committed to luminal or basal lineages, respectively, as early as embryonic day E15.5 in mammary development. This is particularly exciting since all MECs still express K5 (in white in Figs. 2D and EV3E,F) and other known markers of adult LCs and BCs at this developmental stage (Figs. EV1F and EV3G). It is noteworthy that, although mammary buds develop asynchronously, the spatial segregation of the early fate markers *Cxcl14* and *Anxa1* did not differ among different buds (Fig. EV3H).

Considering our findings that a proportion of MECs is already lineage-committed at E15.5, we next sought to examine the spatial localization of cells still possessing a hybrid basal-luminal expression signature within the developing mammary bud. To this aim, we searched for genes associated with the hybrid cell cluster identified at E15.5 (Fig. 1B). A promising candidate marker gene for this cluster was the HLA class II cell surface receptor *Cd74* (Figs. 2C and EV3B), previously proposed as a putative mammary stem cell marker (dos Santos et al, 2013). smRNA-FISH analysis with a probe targeting this gene revealed that, while *Cd74* expression overlapped with both *Anxa1* and *Cxcl14* in early mammary embryonic development (E13.5), a big proportion of *Cd74* transcripts resided in the middle and outer regions of the mammary bud at E15.5, often coinciding with *Cxcl14* expression (Fig. 2G–L). Thus, the hybrid cells we identified by transcriptomic analysis at E15.5 appear to be primarily localized in proximity to the BM, where basal-committed cells are also found within growing mammary buds.

Collectively, our spatial transcriptomic data reveal that the embryonic basal-like and luminal-like mammary cell clusters that we could distinguish by scRNA-seq are found in defined and mostly mutually exclusive positions within the mammary bud at E15.5, at the onset of branching morphogenesis. We believe that spatial segregation and sorting of mammary embryonic progenitors may conceivably underlie their state of differentiation and lineage commitment at this critical stage of embryonic mammary development.

## Identification of two spatially distinct mesenchymal cell populations in the embryonic mammary stroma

Mammary epithelial buds at E13.5 are surrounded by specialized mammary mesenchyme, supporting their subsequent sprouting to invade the underlying fat pat precursor at around E15.5 to initiate the first branching morphogenetic events. Paracrine signaling between mammary epithelial cells and surrounding mesenchymal cells is indispensable for this process (Spina and Cowin, 2021; Wansbury et al, 2011). To gain further insights into mammary mesenchymal patterning during embryonic development, we next focused our analysis on the scRNA-seq data of mesenchymal cells at E13.5, E15.5, and P0. Clustering of non-epithelial cells identified

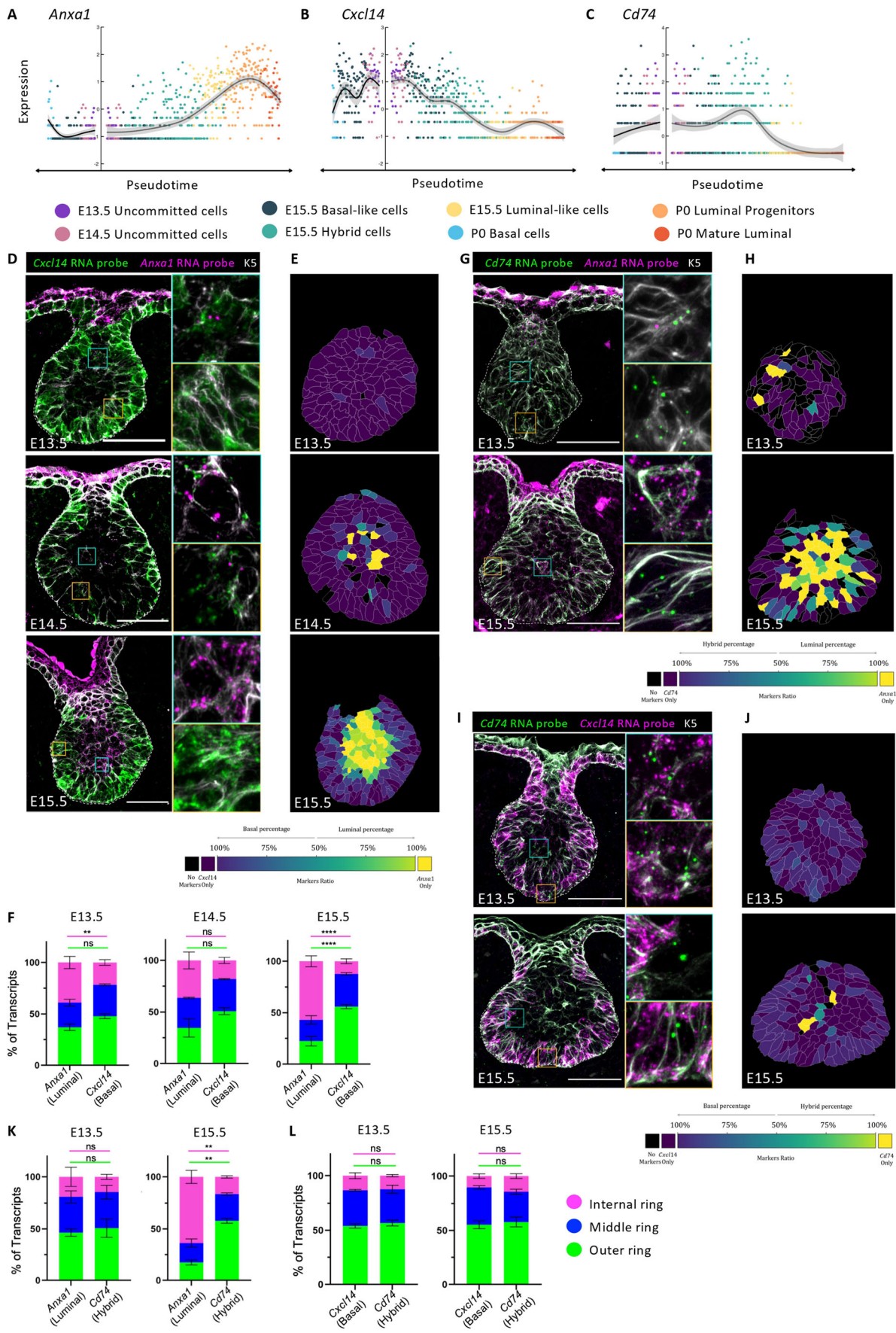

◀ **Figure 2.  Luminal and basal progenitors segregate into different distribution patterns at E15.5.**

(A–C) Characteristic examples of the selected genes presenting increased expression in pseudotime towards luminal differentiation (*Anxa1*, **A**), basal differentiation (*Cxcl14*, **B**) or in the hybrid cells at E15.5 (*Cd74*, **C**). Cells are color-coded by cluster as indicated in the legend. (**D**) Representative sections of embryonic mammary buds at E13.5, E14.5 and E15.5 showing the expression of *Cxcl14* (in green) and *Anxa1* (in magenta) detected by RNAscope and of K5 by IF (in white). A white dotted line delineates the BM (*n* = 3). (**E**) Segmentation of individual mammary cells from (**D**) that were color-coded according to the ratio between the number of transcripts for *Cxcl14* and *Anxa1*, as indicated on the color key at the bottom. The color key goes from dark purple (100% of transcripts in a cell are *Cxcl14*) to bright yellow (100% of transcripts in a cell are *Anxa1*). (**F**) Quantification of the proportion of *Anxa1* and *Cxcl14* transcripts in each ring (see Fig. EV3D) at each developmental stage (*n* = 5). (**G, I**) Representative sections of embryonic mammary buds at E13.5 and E15.5, showing the expression of *Cd74* (in green) and *Anxa1* (in magenta) in (**G**) or *Cxcl14* (in magenta) in (**I**), detected by RNAscope, and of K5 by IF (in white). Dotted lines delineate the BM (*n* = 2). (**H, J**) Segmentation of individual mammary cells from (**G**) and from (**I**) that were color-coded according to the ratio between the number of transcripts for *Cd74* and *Anxa1* in (**H**) or for *Cd74* and *Cxcl14* in (**J**). The color key goes from dark purple (100% of transcripts in a cell are *Cd74* in (**H**) or *Cxcl14* in (**J**)) to bright yellow (100% of transcripts in a cell are *Anxa1* in (**H**) or *Cd74* in (**J**)). (**K, L**) Quantification of the proportion of *Cd74* and *Anxa1* transcripts in (**K**) and of *Cd74* and *Cxcl14* transcripts in (**L**) in each ring at each developmental stage (*n* = 2). Data information: graphs in (**F, K, L**) show mean ± SEM. Statistical significance was assessed with a two-way ANOVA test. ****$P < 0.0001$, **$P < 0.01$, ns non-significant. Scale bars: 50 μm in (**D, G, I**). Source data are available online for this figure.

three mammary mesenchymal cell subsets at each stage (Fig. 3A). By computing a cell cycle score based on a list of cell cycle-related genes, we identified proliferative cell clusters at early developmental timepoints, E13.5 and E15.5 (Figs. 3A and EV4A) (Lan et al, 2023; Myllymäki et al, 2023). Furthermore, immunofluorescence analysis of phosphorylated Histone H3 (PH3) expression at four developmental stages revealed that proliferative mesenchymal cells also exist at later stages (E17.5 and P0) (Fig. EV4B), although they did not to produce a distinguishable and independent UMAP cluster in Fig. 3A.

We next singled out specific markers defining the two non-proliferative mesenchymal clusters at E15.5 (Fig. EV4C). Selected genes included *Esr1* (coding for the Estrogen Receptor ERα), *Plagl1* (coding for the zinc finger protein PLAGL1), *Lef1* (coding for the lymphoid enhancer binding factor 1), *Crabp1* (coding for cellular retinoic acid binding protein 1), *Vcan* (coding for Versican, an ECM proteoglycan) and *Cdkn1c* (coding for Cyclin-dependent kinase inhibitor C1, a negative regulator of the cell cycle), all differentially expressed in opposing mesenchymal clusters (Figs. 3B and EV4C). Immunostaining for ERα and LEF1 showed clear expression in mesenchymal cells surrounding the mammary bud, as previously reported for ERα (Wansbury et al, 2011) (Fig. 3C). Immunofluorescence analysis for PLAGL1, on the other hand, revealed that PLAGL1+ mesenchymal cells are located further away from the mammary epithelium (Fig. 3D). Further analysis of other markers by smRNA-FISH demonstrated a clear enrichment of expression of *Crabp1* and *Vcan* in mesenchymal cells surrounding the embryonic mammary bud, while *Cdkn1c* was enriched in cells found at more distal regions (Fig. 3E). These results indicate that the two transcriptionally distinct mesenchymal populations are also differentially localized within the embryonic mammary stroma at E15.5 and can be categorized based on their proximity to the mammary epithelial bud. We thus refer to cells closest to the epithelium as the sub-epithelial mesenchyme (SE-M) and those located further away as dermal mesenchyme (D-M).

The heterogeneity of mesenchymal cells and the complexity of the mammary stroma increases at birth, where three clusters of fibroblasts can be distinguished, namely *Col15a1+*, *Pi16+*, and *Eln+* clusters, with the first two subsets also identified across 17 other tissues (Buechler et al, 2021). Interestingly, the *Col15a1+* population also expresses *Fabp4*, *Pparg* and *Aoc3*, surface markers of pre-adipocytes. Conversely, the *Pi16+* population of fibroblasts expresses *Dpp4*, *Sema3c*, and *Wnt2*, reported to be upregulated in subcutaneous mesenchymal progenitors (Merrick et al, 2019)

(Figs. 3A and EV4D). Structural and ECM proteins (*Col4a1*, *Col4a2*, *Col18a1*, *Mmp19*, *Sdc1*, *Sparcl1*) are also highly expressed in the *Col15a1+* population. Finally, the *Eln+* mesenchymal population identified at P0 displays elevated expression of *Eln*, *Mfap4*, *Mgp*; genes typically expressed by myofibroblasts.

## FGF10 produced by the dermal mesenchyme is an important regulator of embryonic mammary morphogenesis

Communication between the mammary epithelial and stromal compartments is essential for branching morphogenesis (Inman et al, 2015). Thus, considering the observed spatial patterning of mesenchymal cells at E15.5 (Fig. 3), we next computationally predicted paracrine interactions between the identified mesenchymal cell subsets and MECs using CellPhoneDB (Vento-Tormo et al, 2018). We first focused on ligand–receptor interaction pairs between either the sub-epithelial or dermal mesenchyme and basal-like or luminal-like MECs at E15.5. This approach highlighted several developmental signaling pathways, including FGF, Wnt and TGFβ (Transforming growth factor beta), as putative mediators of the crosstalk between E15.5 LC-like and BC-like cells and the sub-epithelial or dermal mesenchyme (Fig. 4). Specific interactions between TGFβ receptors and their ligand TGFβ2 were highly significant between basal-like and luminal-like MECs and the dermal mesenchymal cells (Fig. 4A). Of interest, the TGFβ3-TGFβR3 interaction pair was instead only significant between basal-like MECs and dermal mesenchymal cells, whereas it was not predicted with LC-like MECs. The WNT pathway is more complex to dissect, since bidirectional signals between epithelium and mesenchyme were predicted, with specific expression of distinct ligands and receptors. For instance, *Fzd4* and *Wnt5a* were expressed in the mesenchymal clusters, with exlusive expression of *Fzd4* in the dermal mesenchyme, while *Wnt3, 4, 7b*, and *10b* signals come from epithelial cells (Fig. 4B). Among the epithelial ligands, *Wnt3* and *Wnt10b* expression was found in basal-like and hybrid cells but not LC-like cells at E15.5 (Figs. EV2C and 4B). This is in line with previous findings reporting that *Wnt10b* is one of the earliest markers localized in the mammary epithelium (Chu et al, 2004).

To address how the ligand-receptor interaction landscape changes before and during the potency switch, we also checked the interactions MECs/stroma at E13.5 and found that most pathways are conserved between E13.5 and E15.5, although specific

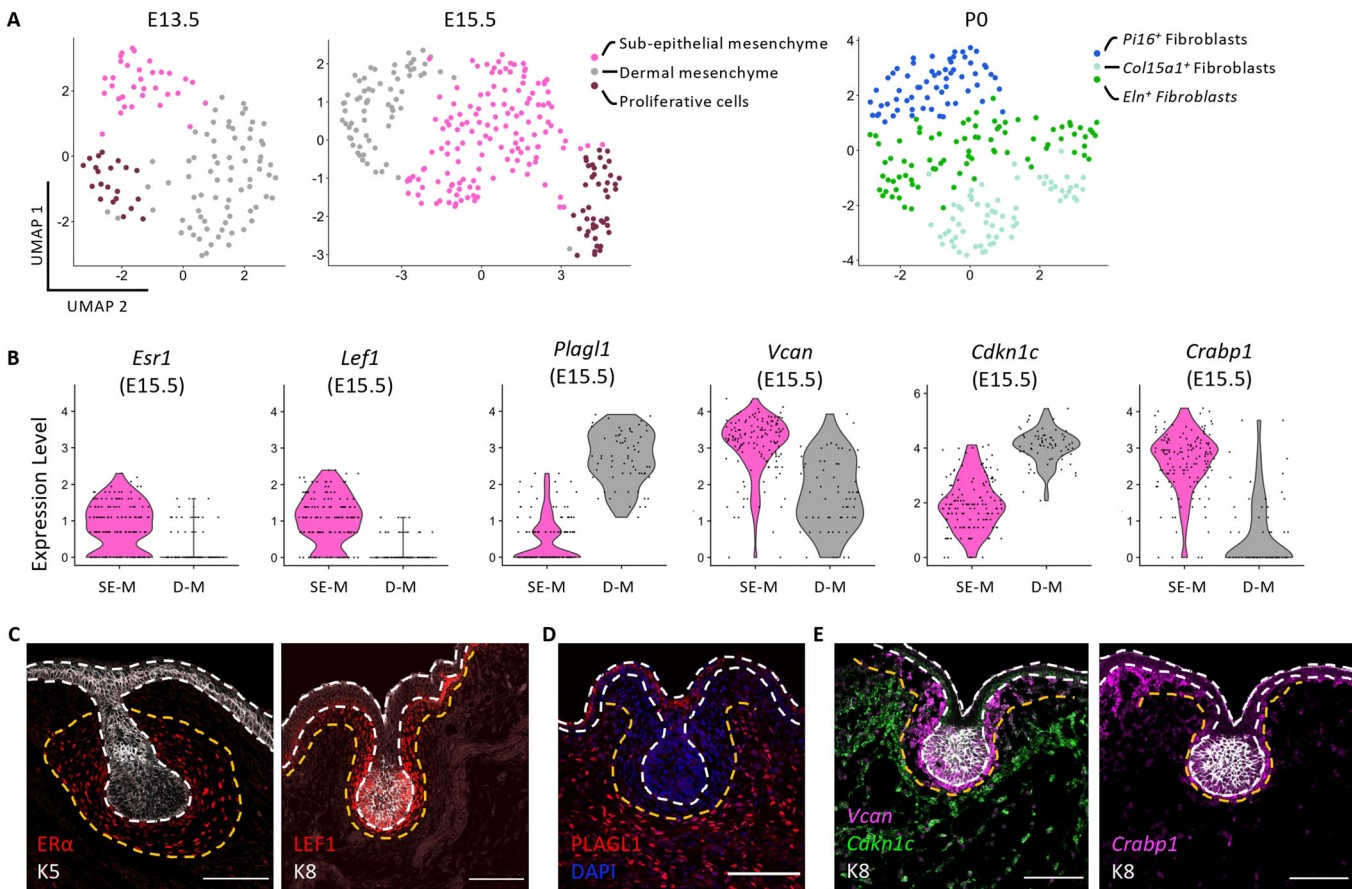

**Figure 3. The early embryonic mammary mesenchyme contains two spatially distinct cell populations.**

(A) UMAP plots of embryonic mammary mesenchymal cells isolated at E13.5, E15.5, and P0. Cells are color-coded by cluster. (B) Violin plots representing the expression levels of *Esr1*, *Lef1*, *Plagl1*, *Vcan*, *Cdkn1c*, and *Crabp1* in sub-epithelial mesenchyme (SE-M) and dermal mesenchyme (D-M) at E15.5 (*n* = 208 cells). (C, D) Representative sections of embryonic mammary buds at E15.5 immunostained for ERα (in red) and K5 (in white) (C) or LEF1 (in red) and K8 (in white) (C); PLAGL1 (in red) and DAPI (in blue) (D). (E) Representative sections of embryonic mammary buds at E15.5 showing the expression of *Vcan* (in magenta), *Cdkn1c* (in green) or *Crabp1* (in magenta), detected by RNAscope and immunostained for K8 (in white). Dotted lines delineate the BM (in white) and the two mesenchymal compartments (in orange). Scale bars: 100 μm (*n* = 3).

differences could be detected (Fig. 4A). For example, although several components of IGF signaling were implicated as important mediators of communication between mesenchyme and MECs at both timepoints, a switch from IGF2 to IGF1-mediated signaling was detected in the E13.5 to E15.5 transition. In addition, the PTHrP signaling pathway is known to govern the transition from the mammary bud to a branching organ. PTHLH, expressed in the epithelium, signals to mesenchymal PTH1R to modulate Wnt and BMP signaling in early mammary development, inducing the production of a specialized condensed mesenchyme that maintains mammary epithelial cell fate (Macias and Hinck, 2012). This PTHLH-PTH1R interaction, scored as highly significant at E13.5, is attenuated as development proceeds, and becomes restricted to the communication between basal-like cells, expressing *Pthlh* and both the sub-epithelial and dermal mesenchyme, equally expressing *Pth1r*. The BMP pathway at work involves the BMP2 and BMP4 ligands, the latest being particularly enriched in the sub-epithelial mesenchyme at E13.5, that signal through BMPR1A to inhibit hair follicle formation in the developing nipple sheath (reviewed in

(Spina and Cowin, 2021)). Other predicted embryonic pathways acting before and after mammary fate commitment include neuropilins (NRP) interacting with semaphorins (SEMA), PDGF, Ephrins and Dlk1 and their respective receptors. Of particular interest, FGF signaling was found to mediate critical cell–cell communication both at E13.5 and E15.5, but differences in ligand expression were observed. Specifically, we focused on the highly significant interaction between the FGF10 ligand, which was found to be exclusively expressed in the dermal mesenchyme, and its receptor FGFR2, expressed in MECs (Fig. 4C). To functionally assess the validity of these computational predictions, we sought to investigate the impact of exogenous FGF10 on embryonic branching morphogenesis, given its reported role in mammary placode development in constitutive FGF10 knock-out mice (Mailleux et al, 2002; Veltmaat et al, 2006) as well as its recently reported role in branch elongation in salivary gland organoids (Kim et al, 2021). We thus tested by live-cell imaging the impact of FGF10 on branching dynamics of mammary buds in ex vivo cultures. Explant cultures provide a highly tractable system for

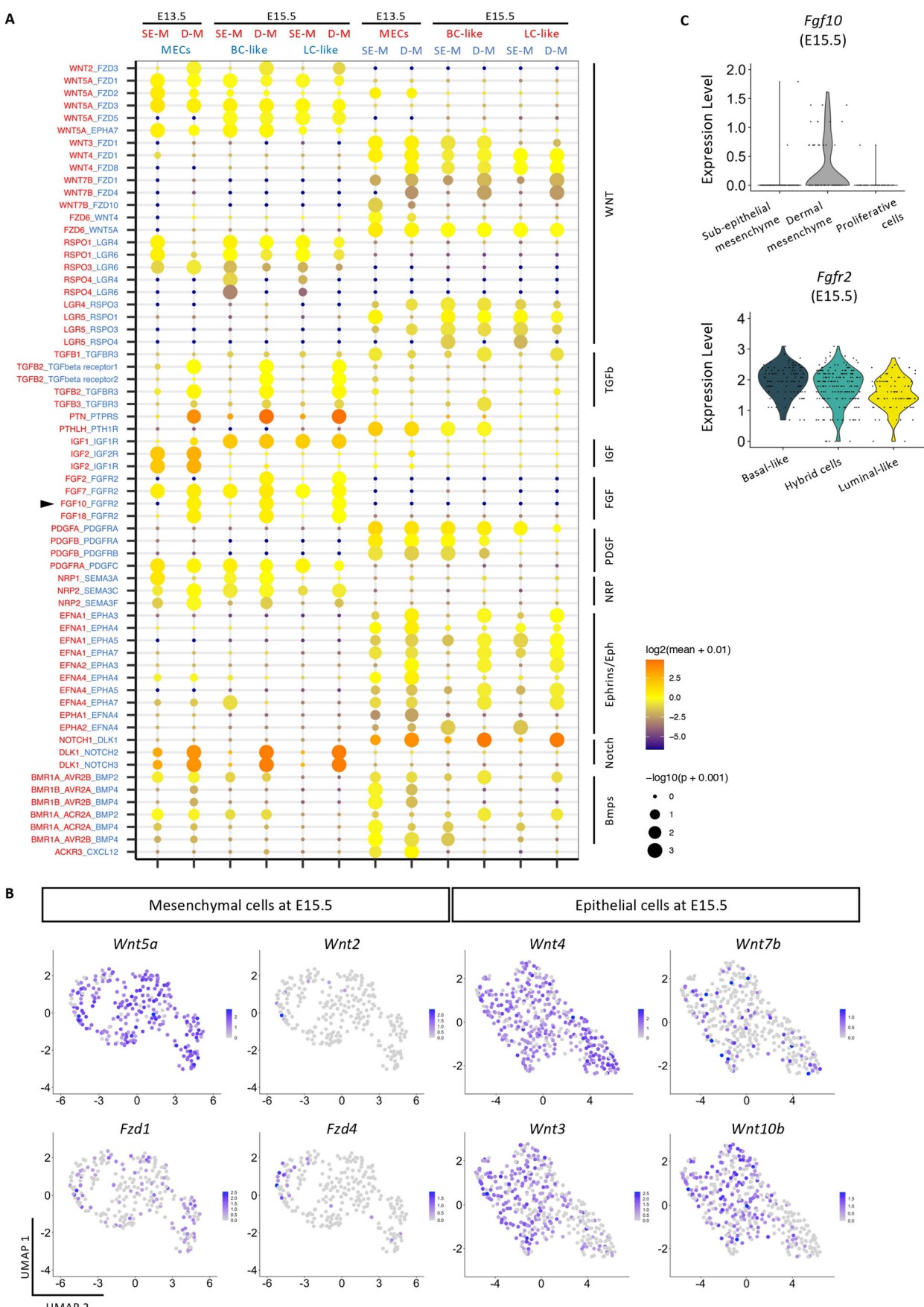

◄  **Figure 4.  Changes in ligand–receptor interaction pairs between both mesenchymal populations and the epithelial cell clusters before and during cell fate switch.**

(A) CellPhoneDB analysis with the predicted ligand–receptor interactions between the two mesenchymal populations, sub-epithelial (SE-M) or dermal mesenchyme (D-M), and mammary epithelial cells at E13.5 and E15.5 (*P* value < 0.01). The arrowhead highlights the ligand–receptor interaction between FGF10 and FGFR2 that was functionally investigated in embryonic ex vivo cultures. Permutation test was used for statistical analysis. (B) UMAP plots from Fig. 3A illustrating the expression of selected Wnt-related cluster-specific genes in mesenchymal cells at E15.5 (left side) and from Fig. 1B illustrating the expression of some cluster-specific Wnt ligands in epithelial cells at E15.5 (right side). (C) Violin plots representing the expression levels of *Fgf10* and *Fgfr2* in mesenchymal and epithelial clusters, respectively, at E15.5 (*n* = 383 cells).

modeling embryonic mammary cell behavior and branching morphogenesis (Carabaña and Lloyd-Lewis, 2022; Voutilainen et al, 2013).

Embryonic mammary buds along with their surrounding mesenchyme were dissected at E13.5 and cultured ex vivo on an air-liquid interface. Embryonic MECs expressed both basal and luminal markers (K5, K14, and P63 for basal cells and K8 for luminal cells) after 24 h in culture (Fig. EV5A–C), consistent with in vivo observations (Fig. EV3G) (Wansbury et al, 2011). During 8 days of ex vivo culture, embryonic mammary buds undergo sprouting and branching, recapitulating the morphogenetic events occurring in vivo (Fig. EV5D,E). Immunostaining of the resulting 8-day-old ductal tree (corresponding to approximately P0/P1 in vivo) revealed that MECs in the outer layer express basal markers such as P63 (Fig. EV5D,EV5F) and α smooth muscle actin (α-SMA) (Fig. EV5E), while inner cells express the luminal marker K8 (Fig. EV5D-E). In addition, polarity acquisition appeared normal, as revealed by apical ZO-1 staining in the internal luminal cells (Fig. EV5F). These results indicate that the fundamental aspects of embryonic mammary morphogenesis and epithelial lineage segregation and differentiation can be reconstituted in ex vivo cultures.

Taking advantage of this powerful system, we next investigated the impact of FGF signaling by undertaking live imaging of embryonic mammary explants cultured with FGF10 (Fig. 5A). To measure the velocity of branch growth in control and FGF10-treated conditions, after 4 days in culture we traced the endpoint of each branch acquired every 60 min for 24 h. By measuring the distance traveled over time in control and FGF10-treated conditions, these experiments indicated that mammary branches grow faster when cultured in the presence of FGF10 (Fig. 5B).

FGF10 secreted by stromal cells may accelerate branching morphogenesis by increasing either epithelial cell proliferation or motility. To discriminate between these two possibilities, we measured the planar surface area of mammary buds over time and found that tissue growth was not significantly affected by FGF10, since the explant area increased two-fold within 16 h of culture in both control and FGF10 conditions (Fig. 5C). While FGF10 is a potent mitogen in several contexts, 5-ethynyl-2′-deoxyuridine (EdU) incorporation experiments suggested that it did not promote mammary epithelial cell proliferation during branch elongation in *ex vivo* cultures (Fig. 5D,E). Moreover, the number of branches in embryonic explant cultures supplemented with FGF10 was equivalent to control cultures (Fig. 5F). However, the diameter of branches at their base was reduced in the presence of FGF10 (Fig. 5G), suggesting that while MECs numbers are equivalent, cells may move faster along extending ducts, which consequently become thinner in the presence of FGF10. Our data therefore shows that, similar to previous observations (Hannezo

et al, 2017; Mailleux et al, 2002; Veltmaat et al, 2006), FGF signaling promotes branching of the embryonic mammary ductal tree at the initial stages of embryonic development, likely by stimulating epithelial cell motility. As the effects of adding exogenous FGF10 to the embryonic mammary gland can be confused by its production by the dermal mesenchyme (Fig. 4C), as well as by exposing both luminal-like and basal-like cells to the same ligand concentrations that may not reflect potential FGF gradients in vivo, we next sought to assess the consequences of disrupting FGFR signaling by treating embryonic mammary gland explants with BGJ398, a potent and highly selective inhibitor of FGFR1/2/3 (Guagnano et al, 2011). E13.5 explants were cultured ex vivo and monitored for the first sprouting events (day 4), at which time the inhibitor was added. Strikingly, mammary gland explants unable to signal through FGFR failed to grow (Fig. 5H), precluding quantification of the velocity of branch elongation during time-lapse imaging. As a result, after a total of 10 days in culture the glands grown in the presence of the FGFR inhibitor covered a significantly smaller planar surface area as compared to controls (Fig. 5I). It is noteworthy that this strong impairment in branching morphogenesis was not accompanied by compromised cell fate specification and lineage segregation, as probed by immunostaining for the lineage markers p63 and K8 in Fig. 5H, suggesting that cell fate commitment does not require branching and might precede ductal morphogenesis.

## Discussion

To generate complex organs of diverse shapes and function, tissue morphogenesis and cell fate specification must be tightly coordinated. Yet, how morphological changes steer individual cells towards a particular fate and, conversely, how cell fate decisions orchestrate morphogenesis, remain ambiguous. By combining temporal scRNA-seq analysis with spatial transcriptomics and live imaging of branching embryonic explant cultures, this work provides original insights into the intrinsic molecular mechanisms as well as non-autonomous positional cues underlying the progressive lineage specification of epithelial cells and concomitant morphogenetic events occurring during embryonic mammary development.

Our data revealed that embryonic MECs at E15.5 can already be distinguished as three transcriptionally discrete cell populations: basal-like, luminal-like and hybrid cells. This was surprising, since previous scRNA-seq studies concluded that bipotent MaSCs, sharing luminal and basal characteristics, exist throughout embryogenesis, and that two separate lineages are only distinguishable postnatally (Giraddi et al, 2018; Wuidart et al, 2018). It is plausible that differences in sequencing depth, FACS-isolation strategies, mouse genetic background, and downstream analysis

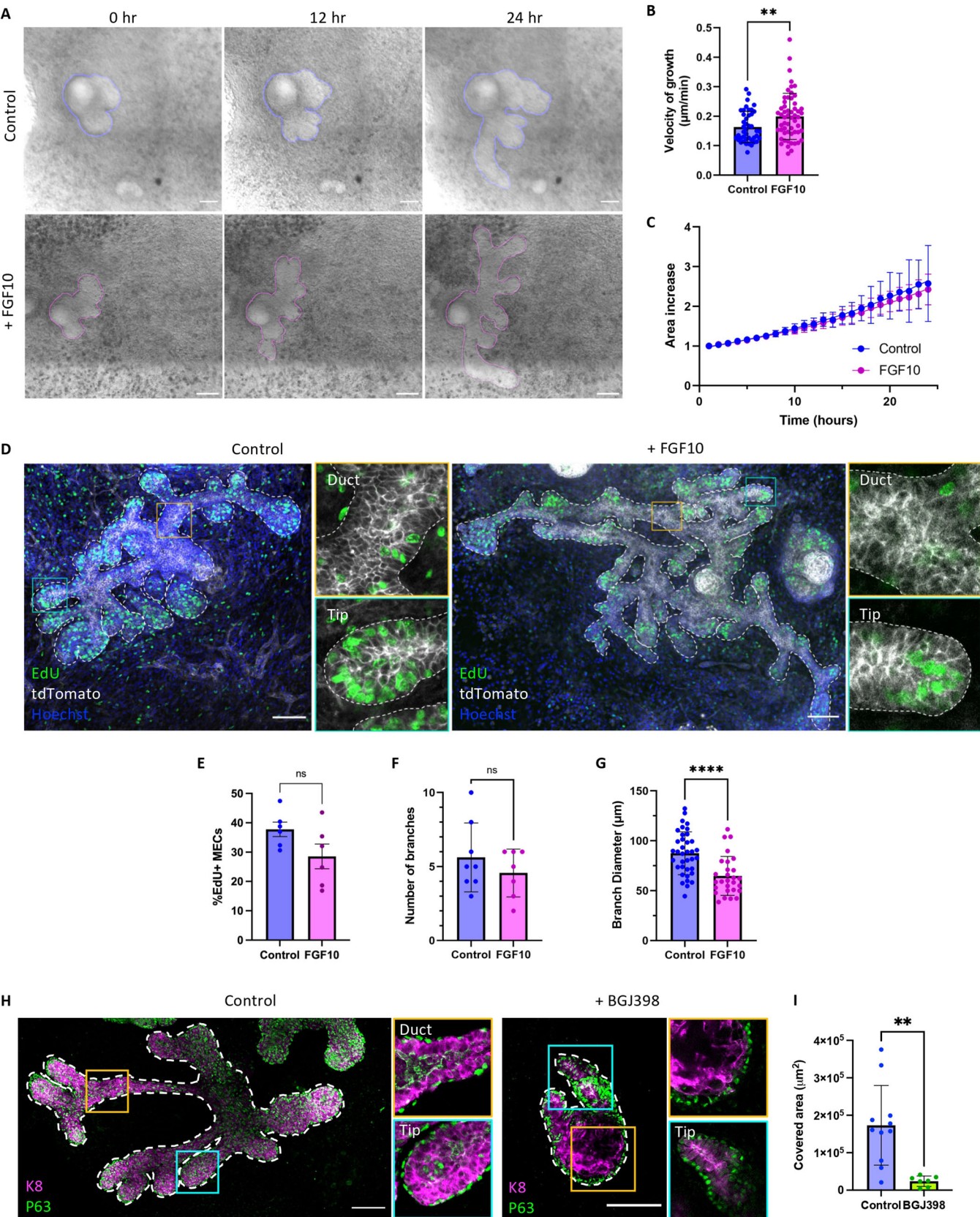

**Figure 5. FGF signaling is required for embryonic mammary branching.**

(A) Time-lapse images of a mammary explant grown in control medium (top) or in the presence of FGF10 (bottom) for 24 h. T = 0 h refers to 4 days in culture. The rendered surface of the mammary epithelium is outlined in blue (in the control bud) and in magenta (in the FGF10 condition). (B) Quantification of the velocity of branch growth in control conditions and in the presence of FGF10 in the medium (n = 43 branches analyzed from 8 independent explants in control conditions and n = 56 branches analyzed from 12 independent explants in FGF10 conditions). (C) Fold change increase in area in control and FGF10 conditions. In both cases, the area is doubled within 16 h in culture. (n = 8 independent explants in control conditions; n = 7 independent explants in FGF10 conditions). (D) Representative whole-mount immunostaining of an embryonic mammary gland cultured in control or FGF10 conditions showing EdU$^+$ cells (in green), membrane tdTomato (in white) and DAPI (in blue). Mammary buds were dissected at day E13.5 and cultured ex vivo for 7 days. Orange outlined insets show a duct region and blue outlined insets show a tip region (n = 3). (E–G) Quantification of EdU$^+$ cells (E), number of branches (F) and branch diameter (G) in control and FGF10 conditions. n = 6 regions analyzed from at least two independent explants per condition in (E); n = at least seven independent explants per condition in (F); n = 37 branches analyzed from eight independent explants in control conditions and n = 28 branches analyzed from seven independent explants in FGF10 conditions in (G). (H) Representative whole-mount immunostaining of an embryonic mammary gland cultured in control conditions (DMSO) or in the presence of a pan-FGFR inhibitor (BGJ398) starting from day 4 in culture, showing the correct positioning of K8$^+$ (in magenta) and P63$^+$ (in green) cells even when branching is abrogated by FGFR inhibition. Orange outlined insets show a duct region and blue outlined insets show a tip region (n = 3). (I) Area quantification in control and BGJ398 inhibitor conditions (n = 11 regions analyzed from three independent explants in control conditions and n = 7 regions analyzed from three independent explants in BGJ398 treated conditions). Data Information: scale bars: 100 μm. (A, D, H) Graphs show mean ± SEM. Statistical significance was assessed with two-tailed unpaired Welch's *t* test. ****P < 0.0001, **P < 0.01, ns non-significant. Source data are available online for this figure.

pipelines may collectively contribute to the variability in findings between different scRNA-seq datasets at similar developmental times. For example, while we FACS-isolated cells based on their expression of EpCAM (Fig. EV1A), Wuidart et al (Wuidart et al, 2018) and Pal et al (Pal et al, 2021) isolated cells based on Lgr5-GFP expression. Moreover, the different downstream analysis pipelines we used likely enabled us to identify distinct embryonic MECs clusters at E15.5 that were previously indistinguishable in other scRNA-seq studies (Giraddi et al, 2018; Pal et al, 2021). Indeed, more recent snATAC-seq analysis of E18.5 and adult MG revealed that E18.5 MECs, although still presenting fetal-specific features, are partially lineage-biased and already harbor adult-like basal, LP and ML characteristics (Chung et al, 2019). The results presented herein are also consistent with our previous lineage-tracing and theoretical modeling analyses (Lilja et al, 2018), which implied that lineage potential restriction coincides with the initiation of branching morphogenesis around E15.5. Collectively, our data supports a model whereby cell fate specification precedes branching morphogenesis. As cells rearrange their position within the growing tissue, coordination between cell differentiation and cell movements may be mediated by their exposure to changing environmental cues. By determining the regional positioning of the different cell clusters that we identified by scRNA-seq, we observed that luminal and basal commitment is indeed reflected by differences in cell localization within the developing mammary epithelium. It is conceivable, therefore, that spatial segregation of mammary embryonic progenitors at this critical stage of development underpins their state of differentiation and lineage commitment.

Based on these results, we propose a dynamic hierarchical model of mammary differentiation spanning embryonic development (Fig. 6A,B). Mammary epithelial cells at E13.5 are undifferentiated and have yet to engage in lineage specification. As development and tissue morphogenesis progress, these putative multipotent embryonic MaSCs will first give rise to basal-like cells, designated as such based on their expression of several genes that define basal mammary cells postnatally. Basal-like cells will then either differentiate into basal unipotent progenitors by P0, or they will transition towards a transcriptionally hybrid state. Hybrid cells, whose lineage potential remain unclear, will gradually lose basal markers and concomitantly acquire luminal traits, eventually giving rise to unipotent luminal cells at birth.

Embryonic MECs co-express the differentiation markers commonly used to distinguish LCs and BCs in the adult mammary gland (Fig. EV3G). This has, to date, hampered studies of the precise timing and molecular regulators of embryonic mammary lineage specification. The comprehensive single-cell transcriptomic atlas compiled in this work enabled the spatial mapping of distinct subsets of embryonic mammary cells, some of which are already committed to basal or luminal fate. In addition to facilitating the in situ identification of potentially multipotent and unipotent mammary progenitors, the lineage-specific genes we discovered may be functionally important for dictating cell fate choices. These novel early markers of luminal or basal commitment likewise provide new specific promoters that could be used in future lineage-tracing studies to definitively establish the differentiation dynamics and lineage potential of early mammary progenitors, and their contribution to postnatal mammary gland development. It should also be noted that some differences in the cellular hierarchy might exist between the mouse mammary gland and the human breast. Indeed, the existence of cells co-expressing some luminal and basal cytokeratins has been reported in the human breast (Dontu and Ince, 2015). The absence of lineage-tracing approaches in the human context, however, makes it impossible to conclude on the potency of these rare cells. Nonetheless, a better understanding of the mechanisms underlying the switch from multipotency to unipotency is imperative and of broad relevance to human biology as cell fate plasticity and reactivation of embryonic multipotency programs contributes to tissue dysfunction and cancer in several epithelial organs (Gupta et al, 2019).

Importantly, our study also provides critical insights into the poorly explored resident mammary embryonic mesenchymal cell populations that direct epithelial branching morphogenesis. We identified specific transcriptional signatures that distinguish two spatially restricted mesenchymal populations in mammary embryonic glands, named sub-epithelial and dermal mesenchyme. It remains unclear however how mesenchymal cells adopt a fibroblast or an adipocyte fate during embryonic development. Addressing this important question awaits future fate-mapping studies using specific stromal Cre drivers based on the promoters of genes identified in this work.

Ligand-receptor pair interaction analysis of the compiled scRNA-seq data implicated several components of the FGF pathway as important mediators of communication between dermal mesenchyme and basal-like cells. While the FGF10/FGFR2 interaction is known to

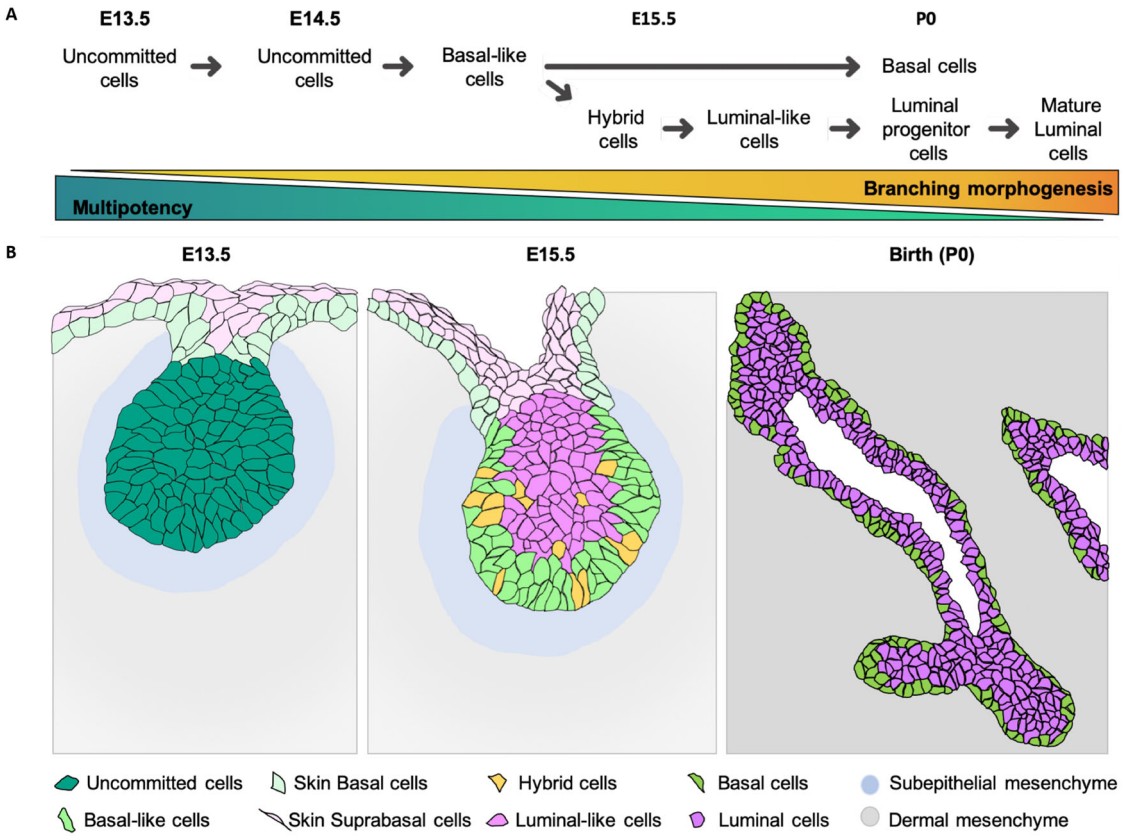

**Figure 6. Proposed model for lineage segregation of embryonic mammary epithelial cells during development.**

(A) Proposed model of luminal and basal differentiation trajectories from E13.5 to P0. (B) Cartoon depicting the spatial distribution of the different cell types distinguishable in the embryonic mammary bud at E13.5, E15.5, and P0.

be a central player in postnatal mammary branching morphogenesis, the importance of this signaling axis on dynamic embryonic mammary branching remained unexplored, limited by the fact that only constitutive knock-out mice for *Fgfr2b* or *Fgf10* existed and they fail to develop mammary placodes, suggesting a requirement to initiate embryonic mammary development (Mailleux et al, 2002). However, this phenotype precluded studies into the role of the FGF10-FGFR2B signaling axis on mammary embryonic development. Thus, to overcome this hurdle and dynamically investigate the effect of FGF10 or FGFR inhibition on embryonic mammary morphogenesis for the first time, we turned to live-cell imaging of explant cultures of mammary embryonic buds that can extensively branch ex vivo, providing opportunities to overcome challenges associated with genetic knock-out models. Our live-imaging data and custom-made image analysis pipeline revealed that, in the presence of exogenous FGF10, embryonic mammary branching is accelerated independently of any effect on cell proliferation or explant growth. On the other hand, pharmacological inhibition of all FGFR signaling completely abrogated branching, recapitulating in vivo phenotypes (Mailleux et al, 2002). It is noteworthy, though, that our studies also provide evidence indicating that impairment of branching elongation did not result in changes in lineage segregation of mammary progenitors, suggesting that cell fate commitment does not require branching. Ideally, to illustrate how MaSCs differentiate according to positional cues, in vivo

cell tracking experiments using fluorescent reporter models (e.g., for *Cxcl14* or *Anxa1* expression) that allow the behavior of genetically labeled embryonic MaSCs to be captured as the tissue develops will be required. This will facilitate studies into the relationship between the initial sprouting events leading to the mammary tree and the time of stem cell potency switch and commitment to a specific lineage.

In summary, this work reveals the heterogeneity in cell states of both the embryonic mammary epithelium and the surrounding mesenchyme and provides important insights into the paracrine cellular interactions that guide branching morphogenesis. Our computational analyses have uncovered the molecular mechanisms and transcription factors involved in regulating mammary cell fate specification. Furthermore, the lineage trajectory analysis reported herein could be extended to other stratified epithelia to determine whether these mechanisms are shared in other epithelial tissues during embryonic development.

## Methods

### Mice

We exclusively analyzed female mice. Ex vivo cultures were established from the double fluorescent reporter R26^mT/mG mice

**Table 1.  Reagents and materials.**

| Reagent | Source | Identifier |
|---|---|---|
| **Chemicals, peptides, and recombinant proteins** | | |
| Triton X-100 | Euromedex | 2000-C |
| Paraformaldehyde | Electron Microscopy Sciences | 15710 |
| Sucrose | Sigma | S0389 |
| DMEM/F-12 | Gibco-Thermo Fisher Scientific | 21331020 |
| Collagenase A | Roche | 10103586001 |
| Hyaluronidase | Sigma-Aldrich | H3884 |
| DNAse I | Sigma-Aldrich | D4527 |
| Aqua-Polymount | Polysciences | 18606 |
| ProLong Diamond Antifade Mountant | Invitrogen-Thermo Fisher Scientific | P36930 |
| Recombinant Mouse FGF10 Protein | Bio-techne | 6224-FG |
| BGJ398 | Selleckchem | S2183 |
| DMSO | Sigma-Aldrich | D2438 |
| Pancreatin from porcine pancreas | Sigma-Aldrich | P3292 |
| Porcine Trypsin | Sigma-Aldrich | 85450C |
| Dispase II | Roche | 04942078001 |
| Ascorbic Acid | Sigma-Aldrich | A4544 |
| GlutaMAX | Gibco-Thermo Fisher Scientific | 35050-038 |
| Fetal bovine serum | Gibco | 10500064 |
| Penicillin–streptomycin | Sigma-Aldrich | P4333 |
| NaCl | Sigma-Aldrich | S5886 |
| KCl | Sigma-Aldrich | P5405 |
| $NaH_2PO_4$ | Sigma-Aldrich | S3522 |
| D-(+)-glucose | Sigma-Aldrich | G7021 |
| $NaHCO_3$ | Sigma-Aldrich | S5761 |
| Tissue-Tek O.C.T. | Sakura | 4583 |
| Quadrol | Sigma-Aldrich | 122262 |
| Urea | Sigma-Aldrich | U5378 |
| 2, 2', 2"-nitrilotriethanol | Sigma-Aldrich | 90279 |
| **Critical commercial assays** | | |
| Click-IT EdU AlexaFluor 647 imaging kit | Invitrogen-Thermo Fisher Scientific | C10640 |
| RNAscope Multiplex Fluorescent Detection Kit v2 kit | ACD | 32310 |
| RNAscope $H_2O_2$ and protease reagents | ACD | 322381 |
| RNAscope Target Retrieval Reagent | ACD | 322000 |
| RNAscope TSA buffer pack | ACD | 322810 |
| RNAscope Probe Diluent | ACD | 300041 |
| TSA PLUS CYANINE 3 | Akoya Biosciences | NEL744001KT |
| TSA PLUS CYANINE 5 | Akoya Biosciences | NEL705A001KT |

| Reagent | Source | Identifier |
|---|---|---|
| TSA PLUS FLUORESCEIN | Akoya Biosciences | NEL741001KT |
| RNAscope Probe- Mm-Anxa1-C2 | ACD | 509291 |
| RNAscope Probe- Mm-Lgals3-C2 | ACD | 461471 |
| RNAscope Probe- Mm-Plet1 | ACD | 557941 |
| RNAscope Probe- Mm-Cxcl14-C3 | ACD | 459741 |
| RNAscope Probe- Mm-Ndnf-C2 | ACD | 447471 |
| RNAscope Probe- Mm-Pthlh-C3 | ACD | 456521 |
| RNAscope Probe- Mm-Cd74 | ACD | 437501 |
| RNAscope Probe- Mm-Vcan | ACD | 486231 |
| RNAscope Probe- Mm-Cdkn1c-C2 | ACD | 458331 |
| RNAscope Probe- Mm-Crabp1-C3 | ACD | 474711 |
| RNAscope 3-plex Positive Control Probe_Mm | ACD | 320881 |
| RNAscope 3-plex Negative Control Probe | ACD | 320871 |
| **Others** | | |
| 35-mm glass bottom dishes | Fluorodish | 81158 |
| Cell culture inserts | Millicell | PICM0RG50 |

(Muzumdar et al, 2007) in a mixed genetic background. Specifically, these mice were initially on a C57 B6/J background, then they were crossed to CD1 and the F1 obtained was then backcrossed once onto a 129 genetic background and then intercrossed. WT C57 B6/N mice were used for the scRNA-seq and RNAscope experiments. All five mammary bud pairs in each embryo were used for RNAscope experiments. Plug detection at mid-day was considered 0.5 days post-coitus (E0.5). Mice were genotyped by PCR on genomic DNA extracted from an earpiece for adult mice or tail tip for embryos. Mouse breeding and husbandry was managed using the mouse colony organization software MiceManager: https://infenx.com/mouse-colony-management-software/.

## Ethics statement

All studies and procedures involving animals were in accordance with the recommendations of the European Community (2010/63/UE) for the Protection of Vertebrate Animals used for Experimental and other Scientific Purposes. Approval was provided by the ethics committee of the French Ministry of Research (reference APAFIS #34364-202112151422480). We comply with internationally established principles of replacement, reduction, and refinement in accordance with the Guide for the Care and Use of Laboratory Animals (NRC 2011). Husbandry, supply of animals, as well as maintenance and care in the Animal Facility of Institut Curie (facility license #C75–05–18) before and during experiments fully satisfied the animal's needs and welfare. All mice were housed and bred in a specific-pathogen-free (SPF) barrier facility with a 12:12 h light–dark cycle and food and water available *ad libitum*.

Mice were sacrificed by cervical dislocation as adults or decapitated as embryos.

## Embryonic mammary gland dissection and ex vivo culture

Mammary embryonic buds were dissected following the protocol developed by the laboratory of M. Mikkola (Voutilainen et al, 2013). Briefly, embryos were harvested from the uterus of a pregnant dam at day E13.5 of pregnancy. Under a dissecting microscope, an incision along the dorsal-lateral line from the hind limb to the forelimb in the right flank of the embryo was done using spring scissors. The flank of the embryo from the incision along the dorsal-lateral line to the midline was detached, and the same steps were repeated for the left flank of the embryo, but this time cutting along the dorsal-lateral line from the forelimb to the hind limb. Tissues were collected in a 24-well plate with phosphate-buffered saline (PBS) until all embryos were dissected.

Next, proteolytic digestion of dissected embryonic flanks was performed as previously described (Lan and Mikkola, 2020). Tissues were incubated with freshly prepared 1.25 U/ml Dispase II solution (Roche, 04942078001) at 4 °C for 15 min. Then, with Pancreatin-Trypsin solution at room temperature (RT) for 4-5 min. To prepare Pancreatin-Trypsin working solution: first, 0.225 g of Trypsin (Sigma-Aldrich, 85450 C) were dissolved into 9 mL of Thyrode's solution [8 g/L NaCl (Sigma-Aldrich, S5886) + 0.2 g/L KCl (Sigma-Aldrich, P5405) + 0.05 g/L NaH$_2$PO$_4$ • H$_2$O (Sigma-Aldrich, S3522) + 1 g/L D-(+)-glucose (Sigma-Aldrich, G7021) + 1 g/L NaHCO$_3$ (Sigma-Aldrich, S5761) dissolved in 1 L of distilled water and filter sterilized]. Then, 1 mL of 10X Pancreatin stock solution [0.85 g NaCl (Sigma-Aldrich, S5886) and 2.5 g Pancreatin (Sigma-Aldrich, P3292) dissolved into 100 mL of distilled water on a magnetic stirrer on ice for 4 h and filter sterilized] and 20 µL of penicillin–streptomycin (10,000 U/ml in stock) (Sigma-Aldrich, P4333) were added. Finally, pH was adjusted to 7.4 with NaOH, and the solution was filter sterilized (see Lan and Mikkola, 2020).

When skin epithelium started to detach from the edges of the mammary mesenchyme, the Pancreatin-Trypsin solution was replaced with DMEM/F-12 (Gibco-Thermo Fisher Scientific, 21331020) embryonic culture medium to inactivate the enzyme activity. After incubating the tissue for 20-30 min in ice, the skin epidermis was removed away from the mesenchyme containing the embryonic mammary buds using two needles.

Mammary embryonic buds were established in ex vivo culture as previously detailed in (Carabaña and Lloyd-Lewis, 2022). Collected embryonic mammary tissue was placed on a cell culture insert floating on embryonic culture medium into a 35-mm cover glass-bottomed tissue culture dish (Fluorodish, 81158). The embryonic culture medium is DMEM/F-12 (Gibco-Thermo Fisher Scientific, 21331020) supplemented with 2 mM GlutaMAX (Gibco-Thermo Fisher Scientific, 35050-038), 10% fetal bovine serum (FBS) (v/v), 20 U/ml Penicillin-Streptomycin (Gibco-Thermo Fisher Scientific, 15140122) and 75 µg/mL Ascorbic acid (Sigma, A4544). Mammary cultures were maintained in a tissue culture incubator at 37 °C with 5% CO$_2$. The culture media was replaced every 2 days. For growth factor assays, 1 nM FGF10 (Bio-techne, 6224-FG) was added to the medium at day 4. For experiments using the FGFR inhibitor, mammary buds were collected from E13.5 female embryos and maintained in culture for 4 days, after which 50 nM BGJ398

(Selleckchem, S2183) or DMSO in control wells were added and renewed every 2 days until day 9 in culture. Complete details of the reagents used are provided in Table 1.

## Mammary cultures whole-mount immunostaining

Whole-mount immunostaining of embryonic tissue explants was performed as previously described (Carabaña and Lloyd-Lewis, 2022). Explants were transferred to 24-well plates, washed in PBS, and fixed with 4% PFA for 2 h at RT. After a blocking step in PBS containing 5% FBS, 1% Bovine Serum Albumin (BSA), and 1% Triton X-100 (Euromedex, 2000-C) for 2 h, explants were incubated with primary antibodies diluted in blocking buffer overnight at 4 °C. Then, with secondary Alexa-fluor conjugated antibodies and DAPI (10 µM) diluted in PBS for 5 h at RT. Ex vivo cultures were mounted in a slide using Aqua-Polymount (Polysciences, 18606). The following primary antibodies were used: rabbit anti-SMA (1:300, Abcam, ab5694), rat anti-K8 (1:300, Developmental Studies Hybridoma Bank, clone TROMA-I), mouse anti-P63 (1:300, Abcam, ab735), rabbit anti-K5 (1:300, Covance, PRB-160P-100), rat anti-ZO-1 (1:100, Millipore, MABT11), rabbit anti-K14 (1:300, ab181595). A complete list of the antibodies used is provided in Table 2.

EdU incorporation was visualized using Click-It chemistry (Invitrogen) by incubating ex vivo cultures for 2 h with EdU solution (10 µM). EdU was then detected with freshly made Click-iT EdU AlexaFluor 647 Imaging Kit (Invitrogen-Thermo Fisher Scientific, C10640), according to the manufacturer's protocol. Nuclei were stained with Hoechst33342 (10 µg/mL) for 30 min at RT.

## Immunostaining on 2D sections

Embryos were harvested and fixed in 4% PFA overnight at 4 °C, followed by another overnight incubation at 4 °C in 30% sucrose. Then, embryos were embedded in an optimum cutting temperature (OCT) compound and 7 µm-thick cryosections were cut using a cryostat (Leica CM1950). After a blocking step in PBS containing 5% FBS, 2% BSA and 0.2% Triton X-100 for 2 h, sections were incubated with primary antibodies diluted in blocking buffer overnight at 4 °C in a humidified chamber, then with secondary Alexa-fluor conjugated antibodies and DAPI (10 µM) diluted in PBS for 2 h at RT. Finally, sections were mounted in a slide using Aqua-Polymount (Polysciences, 18606). The following primary antibodies were used: rat anti-K8 (1:300, Developmental Studies Hybridoma Bank, clone TROMA-I), mouse anti-P63 (1:300, Abcam, ab735), mouse anti-ERalpha (1:20, Agilent-Dako, M7047), rabbit anti-K5 (1:300, Covance, PRB-160P-100), rabbit anti-PLAG1 (1:100) (Spengler et al, 1997) and rabbit anti-LEF1 (1:100, Cell Signaling). Complete details of the antibodies used here are provided in Table 2.

## Optical tissue clearing and whole-mount immunostaining

Mammary glands at E17.5 and at birth were dissected and fixed in 4% PFA overnight at 4 °C. Optical tissue clearing was performed as previously described (Lloyd-Lewis et al, 2016). Briefly, tissues were immersed in CUBIC Reagent 1A (urea (10% w/w), Quadrol (5% w/w), triton X-100 (10% w/w) in distilled water) for 2 days at 37 °C,

washed in PBS and blocked overnight at 4 °C in PBS containing BSA (10%) and triton X-100 (0.5%). Tissues were incubated in primary antibodies diluted in blocking buffer at 4 °C for 3 days with gentle agitation. Then, the samples were washed in PBS three times for 1 h each and incubated with secondary Alexa-fluor conjugated antibodies and DAPI (10 μM) diluted in PBS at 4 °C for 2 days with gentle agitation. Tissues were imaged in CUBIC Reagent 2 (sucrose (50% w/v), thiethanolamine (10% w/v), triton X-100 (0.1% w/v) in distilled water). The following primary antibodies were used: rat anti-K8 (1:300, Developmental Studies Hybridoma Bank, clone TROMA-I), rabbit anti-PH3 (1:300, Millipore). The complete list of all antibodies used is provided in Table 2.

## Single-molecule RNA-fluorescence in situ hybridization (smRNA-FISH)

smRNA-FISH was performed using the RNAscope Multiplex Fluorescent Reagent Kit v2 (Advanced Cell Diagnostics), according to the manufacturer's recommendations. In brief, tissue cryosections were pre-treated with the target retrieval reagent (ACD, 322000) for 5 min and digested with Protease III (ACD, 322381) at 40 °C for 15 min, before hybridization with the target oligonucleotide probes. Probe hybridization, amplification and binding of dye-labeled probes were performed sequentially. For subsequent immunostaining, sections were incubated in blocking buffer (PBS containing 5% FBS and 2% BSA) for 1 h. For smRNA-FISH in ex vivo explant cultures, the blocking buffer also included 0,3% Triton X-100 (Euromedex, 2000-C) to allow tissue permeabilization. Incubation with primary antibodies diluted in blocking buffer was performed overnight at 4 °C in a humidified chamber, then secondary antibodies and DAPI diluted in PBS were added for 2 h at RT. The experiments were performed on at least three different embryos for each probe. Slides were mounted in ProLong Diamond Antifade Mountant (Invitrogen-Thermo Fisher Scientific, P36930) for imaging. The following RNAscope probes were used: Mm-Anxa1-C2 (ACD, 509291), Mm-Lgals3-C2 (ACD, 461471), Mm-Plet1-C1 (ACD, 557941), Mm-Ly6d-C1 (ACD, 532071), Mm-Cxcl14-C3 (ACD, 459741), Mm-Ndnf-C2 (ACD, 447471), Mm-Pthlh-C3 (ACD, 456521), Mm-Cd74-C1 (ACD, 437501), Mm-Vcan (ACD, 486231), Mm-Cdkn1c-C2 (ACD, 458331), Mm-Crabp1-C3 (ACD, 474711), 3-plex Positive Control Probe-Mm (ACD, PN 320881) and 3-plex Negative Control Probe (ACD, PN 320891). The complete list of RNAscope probes used is provided in Table 1.

## Microscopy and image acquisition

### 3D imaging

Images were acquired using a LSM780 or LSM880 inverted laser scanning confocal microscope (Carl Zeiss) equipped with 25×/0.8 OIL LD LCI PL APO or 40×/1.3 OIL DICII PL APO. For standard 4-color imaging, laser power and gain were adjusted manually to give optimal fluorescence for each fluorophore with minimal photobleaching. Images were captured using the ZEN Imaging Software and processed in Fiji (ImageJ v1.53).

### smRNA-FISH

Images were acquired using a LSM880 confocal microscope with an Airyscan system. The Airyscan system has 32-channel GaAsP (Gallium Arsenide Phosphide) detectors, which allow to obtain images with enhanced spatial resolution and improved signal-to-noise ratio (SNR) than in traditional LSM systems (Huff, 2015). A 63×/1.4 OIL DICII PL APO objective was used. Images were processed in Fiji (ImageJ v1.53).

### Live imaging

Time-lapse images were acquired using an LSM780 or LSM880 inverted laser scanning confocal microscope (Carl Zeiss) equipped with 10×/0.3 DICI EC PL NEOFLUAR, for imaging at the tissue scale. Explants were cultured in a humidified chamber at 37 °C with 5% $CO_2$ during imaging. To analyze branching morphogenesis in embryonic mammary buds, images were acquired at 8 mm Z intervals over ~80 mm thickness and 60 min intervals for 12-48 h.

## Single-cell dissociation of embryonic mammary gland

The isolated embryonic mammary rudiments include both the mammary epithelium and the surrounding mesenchyme. In total, 60-90 mammary rudiments were dissected for each experiment from 7 to 12 female embryos derived from 2 to 4 timed pregnant females. The scRNA-seq of each developmental time was performed in a separate dissection session to maximize the number of mammary buds analyzed/timepoint. All five mammary bud pairs in each embryo were pooled for scRNA-seq and were equally represented at each developmental timepoint.

Embryonic mammary buds along with their surrounding mesenchyme were dissected as detailed above (see "Embryonic mammary gland dissection and ex vivo culture"). Single-cell dissociation was performed as previously described (Wuidart et al, 2018) with the following modifications:

For mammary rudiments at E13.5, E14.5, and E15.5, single-cell dissociation was performed through enzymatic digestion with 300 U/ml collagenase A (Roche, 10103586001) and 300 U/ml hyaluronidase (Sigma, H3884) for 90 min at 37 °C under shaking. Mammary rudiments from each female embryo were dissociated in a separated 2 mL protein LoBind tube (Eppendorf, 022431102). Cells were further treated with 0.1 mg/ml DNase I (Sigma, D4527) for 3 min. 10% FBS diluted in PBS was added to quench the DNase I. Cells were pelleted by centrifugation at 320 ×g for 10 min.

For mammary glands at birth, the enzymatic digestion for single-cell dissociation was optimized as follows: enzymatic tissue digestion was performed with 600 U/ml of collagenase A (Roche, 10103586001) and 150 U/ml of hyaluronidase (Sigma, H3884) for 90 min at 37 °C on shaking. Cells were further treated with 0.1 mg/ml DNase I (Sigma, D4527) for 3 min and an additional incubation in 0.63% $NH_4Cl$ for 1 min allowed lysis of red blood cells. Cells were pelleted by centrifugation at 320× g for 10 min.

For all developmental times, after careful removal of the supernatant, cells were incubated in fluorescently labeled primary antibodies.

## Cell labeling, flow cytometry, and sorting

Single-cell suspensions were incubated for 15 min on ice with fluorescently labeled primary antibodies diluted in HBSS with 2% FBS. Cells were washed from unbound antibodies with 2% FBS in HBSS and the cell suspension was filtered through a 40-μm cell

**Table 2. Antibodies.**

| Reagent | Source | Identifier |
|---|---|---|
| **Antibodies** | | |
| Rabbit anti-SMA | Abcam | Cat# ab5694; RRID: AB_2223021 |
| Rat anti-K8 | Developmental Studies Hybridoma Bank, University of Iowa | Cat# TROMA-I: RRID: AB_531826 |
| Mouse anti-p63 | Abcam | Cat# ab735; RRID:AB_305870 |
| Rabbit anti-K5 | Covance | Cat# PRB-160P-100; RRID:AB_291581 |
| Rabbit anti-K14 | Abcam | Cat# ab181595, RRID:AB_2811031 |
| Mouse anti-ERalpha | Agilent-Dako | Cat# M7047, RRID:AB_2101946 |
| Rat anti-ZO-1 | Millipore | Cat# MABT11, RRID:AB_10616098 |
| Rabbit anti-PLAG1 | (Spengler et al, 1997) | N/A |
| Rabbit anti-Lef1 | Cell Signaling | Cat #2230, RRID:AB_823558 |
| Rabbit anti-PH3 | Millipore | Cat #06-570, RRID:AB_310177 |
| Goat anti-rabbit AlexaFluor-coupled to different fluorochromes (Cy3, Cy5, A488) | Invitrogen-Thermo Fisher Scientific | Cat# A10520; RRID:AB_2534029 Cat# A10523; RRID:AB_2534032, Cat# A-11034; RRID:AB_2576217 |
| Goat anti-rat AlexaFluor-coupled to different fluorochromes (Cy3, Cy5, A488) | Invitrogen-Thermo Fisher Scientific | Cat# A10522; RRID:AB_2534031, Cat# A10525; RRID:AB_2534034, Cat# A-11006; RRID:AB_2534074 |
| Goat anti-mouse AlexaFluor-coupled to different fluorochromes (Cy3, Cy5, A488) | Invitrogen-Thermo Fisher Scientific | Cat# A10521; RRID:AB_2534030, Cat# A10524; RRID:AB_2534033 Cat# A-11001; RRID:AB_2534069 |
| Goat anti-chicken AlexaFluor- 488 | Invitrogen-Thermo Fisher Scientific | Cat# 400612, RRID:AB_326556 |
| PE anti-mouse Epcam | Biolegend | Cat# 118206, RRID:AB_1134176 |
| APC/Cy7 anti-mouse CD49f | Biolegend | Cat# 313628; RRID:AB_2616784 |
| APC anti-mouse CD31 | Biolegend | Cat# 102510; RRID:AB_312905 |
| APC anti-mouse Ter119 | Biolegend | Cat# 116212; RRID:AB_313713 |
| APC anti-mouse CD45 | Biolegend | Cat# 103112, RRID:AB_312977 |
| PE rat IgM | Biolegend | Cat# 400808; RRID:AB_326584 |
| APC/Cy7 rat IgG2a | Biolegend | Cat# 400524 |
| APC rat IgG2b | Biolegend | Cat# 400612, RRID:AB_326556 |

strainer to eliminate cell clumps. Cell viability was determined with DAPI, and doublets were systematically excluded during analysis. CD45[+], CD31[+], Ter119[+] (Lin[+]) non-epithelial cells were excluded. FACS analysis was performed using an ARIA flow cytometer (BD).

The following primary antibodies were used at a 1:100 dilution: APC anti-mouse CD31 (Biolegend, 102510), APC anti-mouse Ter119 (Biolegend, 116212), APC anti-mouse CD45 (Biolegend, 103112), APC/Cy7 anti-mouse CD49f (Biolegend, 313628), and PE anti-mouse EpCAM (Biolegend, 118206). The isotype controls were the following: PE rat IgM (Biolegend, 400808), PE/Cy7 rat IgG2a (Biolegend, 400522), APC/Cy7 rat IgG2a (Biolegend, 400524) and APC rat IgG2b (Biolegend, 400612). Complete details of the antibodies used are provided in Table 2. The results were analyzed using the FlowJo software (V10.0.7).

## Image analysis and quantification

For time-lapse live-imaging analysis, first time-lapse reconstructions were generated using the Bio-Formats plugin (Linkert et al, 2010) in Fiji (ImageJ v1.53). Then, automated segmentation of mammary buds was performed using a custom-made segmentation model based on U-Net (Ronneberger et al, 2015). Segmented masks and raw image were input in the ImageJ plugin, BTrack, for tracking the growing branch tips. BTrack allows the users to remove or create new end points to manually correct the obtained tracks. We obtained the average growth rate for each branch using customized Python scripts (see "Software availability"). Statistical analyses were performed in Prism (v9.2, GraphPad).

To determine bud surface area in the presence of FGF10 in the medium, segmented masks were obtained from each timepoint using the U-Net model previously described. Generated masks were manually checked and corrected against raw data for consistency prior to extracting area measurements. Surface area was measure for each timepoint and statistical analyses were performed with Prism (v9.2, GraphPad).

Quantification of the smRNA-FISH dots was performed using Python (Python 3.9.13). A custom napari plugin (napari-bud-cell-segmenter) was developed to draw the outline of the mammary buds, perform single-cell segmentation, detect each transcript as a unique fluorescent dot and extract quantitative metrics per cell. The

2D color-coded spatial distribution of the transcripts were obtained using customized Python scripts. First, a mask delimitating the outline of the bud was manually created using our napari plugin and was used for distance map computation. Second, segmentation of the MECs within the mask was performed on the K5 membrane staining channel using watershed segmentation. Finally, smRNA-FISH dot detection was performed for the *Cxcl14*, *Anxa1*, and *Cd74* RNA probes using Laplacian of Gaussian on the normalized corresponding channels with a fixed minimum sigma of 1, maximum sigma of 2 and an adapted thresholding value. Both segmentation and transcripts detection were visually inspected for accuracy and corrected if needed. The number of detected transcripts per segmented cell was computed and the ratio of the different types of detected transcripts was mapped on a custom color key to reveal the distribution of the transcript types per segmented cell.

A custom ImageJ macro was coded to create three parallel regions of interest (ROIs) with a ring-shaped surface (outer, middle, and internal ring). The number of dots in each ROI was calculated for every smRNA-FISH probe using the Find Maxima tool in Fiji (ImageJ v1.53). The percentage of dots in each ring was calculated as the ratio of a number of dots in each ROI over total number of dots in the three ROIs. Statistical analysis was executed in Prism v9.2, GraphPad.

For EdU quantification, two to three independent explants in each condition were analyzed. For each explant, independent regions of interest were randomly selected in discrete Z-slides. The mammary epithelium was outlined manually in Fiji using the tdTomato or luminal lineage marker staining as a guide (ImageJ v1.53). Hoechst images were processed with a median filter (1-2px). StarDist (Schmidt et al, 2018; Weigert et al, 2020) was used to segment and quantify a number of Nuclei and EdU⁺ nuclei within the outlined mammary epithelial tree region in Fiji (ImageJ v1.53). EdU⁺ nuclei were expressed as a percentage of the total number of nuclei. Statistical analysis was performed in Prism (v9.2, GraphPad).

## scRNA-seq data processing and cluster analysis

Single-cell capture and library construction were performed using the 10x Genomics Chromium Single Cell 3' v3.1 kit according to the manufacturer's instructions for samples of all developmental stages. The libraries were sequenced with an Illumina NovaSeq 6000 sequencer by the *Next Generation Sequencing* platform of Institut Curie.

### Data pre-processing and quality control
The 10x Genomics Cell Ranger Single-Cell Software Suite was used for demultiplexing, read alignment, and unique molecular identifier (UMI) quantification (http://software.10xgenomics.com/single-cell/overview/welcome). The pre-built mm10 reference genome obtained from the 10X Genomics website was used to align the reads. The count matrices were individually loaded for each sample in R and analyzed using the Seurat package v4.0.5 (Hao et al, 2021).

Genes expressed in less than three cells and cells with UMI count <5000 and mitochondrial UMI count >6% were removed. This resulted in the following total number of high-quality cells: 228 at E13.5, 59 at E14.5, 740 at E15.5, 409 at P0 in WT mice.

### Normalization
Objects were normalized separately using the SCTransform method, implemented in the "SCTransform" function from Seurat. Briefly, this method regresses out the sequencing depth variation between cells using a negative binomial regression model with regularized parameters (Hafemeister and Satija, 2019).

### scRNA-seq data dimension reduction and clustering
Principal Component Analysis (PCA) was performed on the top 2000 highly variable genes of the SCT assay from the "SCTransform" step. The top 15 principal components (PCs) were further selected (based on inspection of PC elbow plot) to perform graph-based clustering and cell cluster detection. All the Uniform Manifold Approximation and Projection (UMAP) plots (McInnes et al, 2018) were computed using the "RunUMAP" Seurat function with default Seurat parameters.

### Cell cluster identification
Cell clustering was performed using a two-step wise approach, using the "FindNeighbors" and "FindClusters" functions, respectively. The "FindClusters" function was used to set the resolution parameter to 0.8.

### Differential expression analysis
Cell-type marker genes for each cluster were identified using the function "FindAllMarkers" function in Seurat, with detected in minimum cell fraction >10% and log-fold change >0.1. Then, cell clusters were manually annotated based on cell-type-specific markers known to be enriched in each cell population. Cell proliferative clusters were identified using the following list of genes: 'Pclaf', 'Ncapg2', 'Smc2', 'Tyms', 'Tuba1b', 'Hmgb2', 'Top2a', 'Tacc3', 'Cenph', 'Cdk1', 'Tubb5', 'Diaph3', 'Cenpf', in order to compute an expression score using the Seurat function 'AddModuleScore'.

### Signature construction
A single-cell ID score for "basal-like" and "luminal-like" cells was calculated based on previously published transcriptomic analyses of adult MECs (Kendrick et al, 2008). The scores were computed using the Seurat function "AddModuleScore".

### 3D trajectory and pseudotime analysis
For data integration and analysis, only epithelial cell clusters across E13.5, E14.5, E15.5, and P0 were considered. The pre-processing steps previously described were re-applied (normalization, PCA, and basal and luminal score). Epithelial cells were then mapped in a 3D space including the luminal score and basal score on the x axis and the PC related to developmental time on the y axis. For each cell cluster, the coordinates of the center in the 3D space with the median for each dimension were calculated and called "pseudo-bulks". A minimum spanning tree (MST) was generated to connect all pseudo-bulks. Basal and luminal trajectories were inferred through the MST.

To obtain the pseudotime of each cell along the basal or luminal trajectories, each cell was projected in the 3D space to the basal and luminal trajectories separately. Then, the pseudotime for each cell was defined as their distance from the initial point of the trajectory.

The luminal and basal gene expression heatmaps were generated on the pseudotime using the "pheatmap" package. Briefly, the genes

with the top 10% variation across cells within a lineage were selected. The gene expression values were smoothed versus the pseudotime using the generalized additive model (GAM). The hierarchical gene clusters were generated with Euclidean distance and Complete clustering algorithm.

### Cell-cell interaction analysis

The cell-cell interaction analysis was performed using the CellPhoneDB version 3.0.0 (Vento-Tormo et al, 2018) with a *P* value threshold of 0.01. The CellPhoneDB database is publicly available at https://www.cellphonedb.org/. It is a repository of curated receptors, ligands and their interactions that allow to predict potential cell-cell communication mechanisms in single-cell transcriptomic data. For statistical analysis, CellphoneDB uses empirical shuffling to calculate which ligand-receptor pairs display significant cell-type specificity. Specifically, it estimates a null distribution of the mean of the average ligand and receptor expression in the interacting clusters by randomly permuting the cluster labels of all cells. The *P* value for the likelihood of cell-type specificity of a given receptor-ligand complex is calculated on the basis of the proportion of the means that are as high as or higher than the actual mean.

### Statistics and reproducibility

Animals were randomized and analyzed in a non-blinded manner. Each mouse in every experiment was reported on, no mice were excluded except for males. No statistical method was used to predetermine group size. The number of biological replicates was estimated based on the results heterogeneity observed in pre-liminary experiments. At least $n = 2$ animals were used for each experiment, and experiments with at least $n = 3$ replicates were used to calculate the statistical significance of each analysis. Statistical tests and further graphs were prepared in Prism (v9, GraphPad). All graphs show mean ± SEM. Differences between groups were assessed with two-tailed unpaired *T* test with Welch's correction. Statistical analyses between the localization of two RNA probes were assessed with two-way ANOVA test. The significance threshold was $P < 0.05$. *$P < 0.05$, **$P < 0.01$, ***$P < 0.001$, and ****$P < 0.0001$.

### Software availability

Customized scripts and instructions are available from our Github page: https://github.com/Fre-Team-Curie/Embryo-mammary-gland. The web application tool to explore our data and facilitate 3D trajectory analysis visualization throughout our developmental atlas is publicly available open access at: https://sunwjie.shinyapps.io/Embryo_scRNASeq/.

## Data availability

The single-cell RNA sequencing data have been deposited in NCBI's Gene Expression Omnibus (GEO) repository and are accessible through GEO Series accession number GSE210594. Raw Imaging data have been deposited on the BioImage Archive, Accession Number: S-BIAD1099. All data supporting the conclusions of this study are provided in the main text or supplementary

materials. The source data of this paper are collected in the following database record: biostudies:S-SCDT-10_1038-S44318-024-00115-3.

## Peer review information

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

## Acknowledgements

The authors thank S. Tajbakhsh for sharing the mTmG reporter line. We wish to warmly acknowledge the flow cytometry and cell sorting platform at Institute Curie for their technical support, the in vivo experimental facility for help in the maintenance and care of our mouse colony, the ICGex NGS platform of Institut Curie (supported by grants ANR-10-EQPX-03, Equipex and ANR-10-INBS-09-08, France Génomique Consortium, from the Agence Nationale de la Recherche "Investissements d'Avenir" program, by the Canceropole Ile-de-France and by the SiRIC-Curie program - SiRIC Grant INCa-DGOS- 4654), and the Cell and Tissue Imaging Platform-PICT-IBiSA at Institut

Curie (member of the French National Research Infrastructure France-Bioimaging, ANR-10-INBS-04) for their expertise and help. The authors especially thank Olivier Leroy and Anne-Sophie Mace for help with image analysis. We are grateful to the other members of the Fre laboratory for constructive discussions. This work was supported by Paris Sciences et Lettres (PSL* Research University) (grant # C19-64-2019-228), the French National Research Agency (ANR) grant numbers ANR-15-CE13-0013-01 and ANR-21-CE13-0047, the Medical Research Foundation FRM "FRM Equipes" EQU201903007821, the FSER (Fondation Schlumberger pour l'éducation et la recherche) FSER20200211117, the Association for Research against Cancer (ARC) label ARCPGA2021120004232_4874 and by Labex DEEP ANR-Number 11-LBX-0044 to SF. CC was funded by the European Union's Horizon 2020 research and innovation program under the Marie Skłodowska-Curie grant agreement No. 666003 for PhD fellowships and the Medical Research Foundation FRM Project 12917. BL-L is funded by a Vice-Chancellor's Research Fellowship from the University of Bristol and acknowledges support from the Academy of Medical Sciences/Wellcome Trust/the Government Department of Business, Energy and Industrial Strategy/British Heart Foundation/Diabetes UK Springboard Award [SBF003/1170], Elizabeth Blackwell Institute for Health Research (University of Bristol), and Wellcome Trust Institutional Strategic Support Fund (204813/Z/16/Z). CVR was supported by a Medical Research Foundation FRM post-doctoral fellowship (SPF202309017673). The funders had no role in study design, data collection and analysis, decision to publish, or preparation of the manuscript.

## Author contributions

**Claudia Carabaña**: Conceptualization; Data curation; Formal analysis; Validation; Investigation; Methodology; Writing—original draft; Writing—review and editing. **Wenjie Sun**: Resources; Data curation; Software; Formal analysis; Methodology. **Camila Veludo Ramos**: Data curation; Formal analysis; Validation; Investigation; Methodology. **Mathilde Huyghe**: Methodology. **Meghan Perkins**: Methodology. **Aurélien Maillot**: Resources; Data curation; Software; Formal analysis; Methodology. **Robin Journot**: Resources; Data curation; Software; Formal analysis; Methodology. **Fatima Hartani**: Methodology. **Marisa M Faraldo**: Data curation; Formal analysis; Methodology; Writing—original draft; Writing—review and editing. **Bethan Lloyd-Lewis**: Conceptualization; Data curation; Formal analysis; Supervision; Validation; Methodology; Writing—original draft; Writing—review and editing. **Silvia Fre**: Conceptualization; Data curation; Formal analysis; Supervision; Funding acquisition; Validation; Writing—original draft; Project administration; Writing—review and editing.

Source data underlying figure panels in this paper may have individual authorship assigned. Where available, figure panel/source data authorship is listed in the following database record: biostudies:S-SCDT-10_1038-S44318-024-00115-3.

## Disclosure and competing interests statement

The authors declare no competing interests.

# Expanded View Figures

**Figure EV1.   Related to Fig. 1. Lineage-committed cells exist in early MG development.**

(**A**) Representative FACS dot plots of the gating strategy used to sort E15.5 and P0 epithelial and mesenchymal cells. (**B**) UMAP plots of embryonic MECs and surrounding mesenchymal cells isolated by scRNA-seq at E13.5, E14.5, E15.5 and P0. Cells are color-coded by cluster. (**C**) UMAP plot of embryonic MECs isolated at E15.5 after subset analysis of all MECs (including proliferative cells shown in light blue). (**D**) Violin plot representation of the cell cycle score in each mammary epithelial cluster at E15.5 ($n = 430$ cells analyzed). (**E**) Heatmap showing the expression of genes specific for each cell cluster at E15.5. Each column is color-coded according to the cell cluster from (**B**). (**F**) UMAP plots from (**C**) showing the expression of specific luminal (*Krt8* and *Krt18*) and basal (*Krt5* and *Trp63*) genes commonly used to distinguish adult LCs and BCs but unable to discriminate distinct cell clusters at E15.5. (**G**) Box plots illustrating the log2 fold change of the luminal/basal score ratio in each cluster. $n = 22$ cells at E13.5; $n = 28$ cells at E14.5; $n = 98$ basal-like cells, $n = 199$ hybrid cells and $n = 86$ luminal-like cells at E15.5; $n = 19$ basal cells, $n = 140$ LP cells and $n = 39$ ML cells at P0. Statistical significance was assessed with Wilcoxon test. Lower and upper hinges correspond to the first and third quartiles. The upper whisker extends from the hinge to the largest value no further than 1.5 * IQR from the hinge (where IQR is the inter-quartile range, or distance between the first and third quartiles). The lower whisker extends from the hinge to the smallest value at most 1.5 * IQR of the hinge. Data beyond the end of the whiskers are called "outlying" points and are plotted individually.

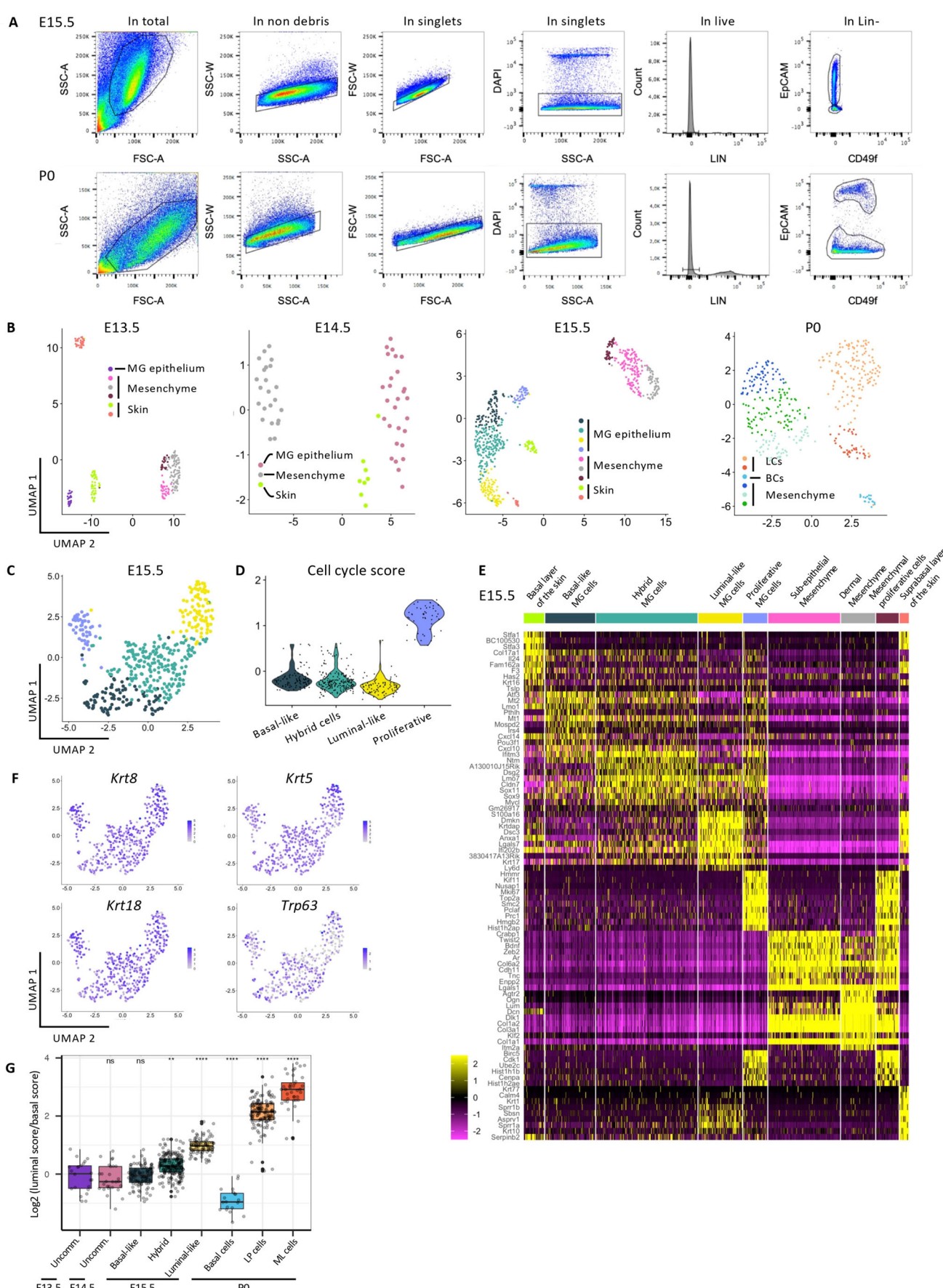

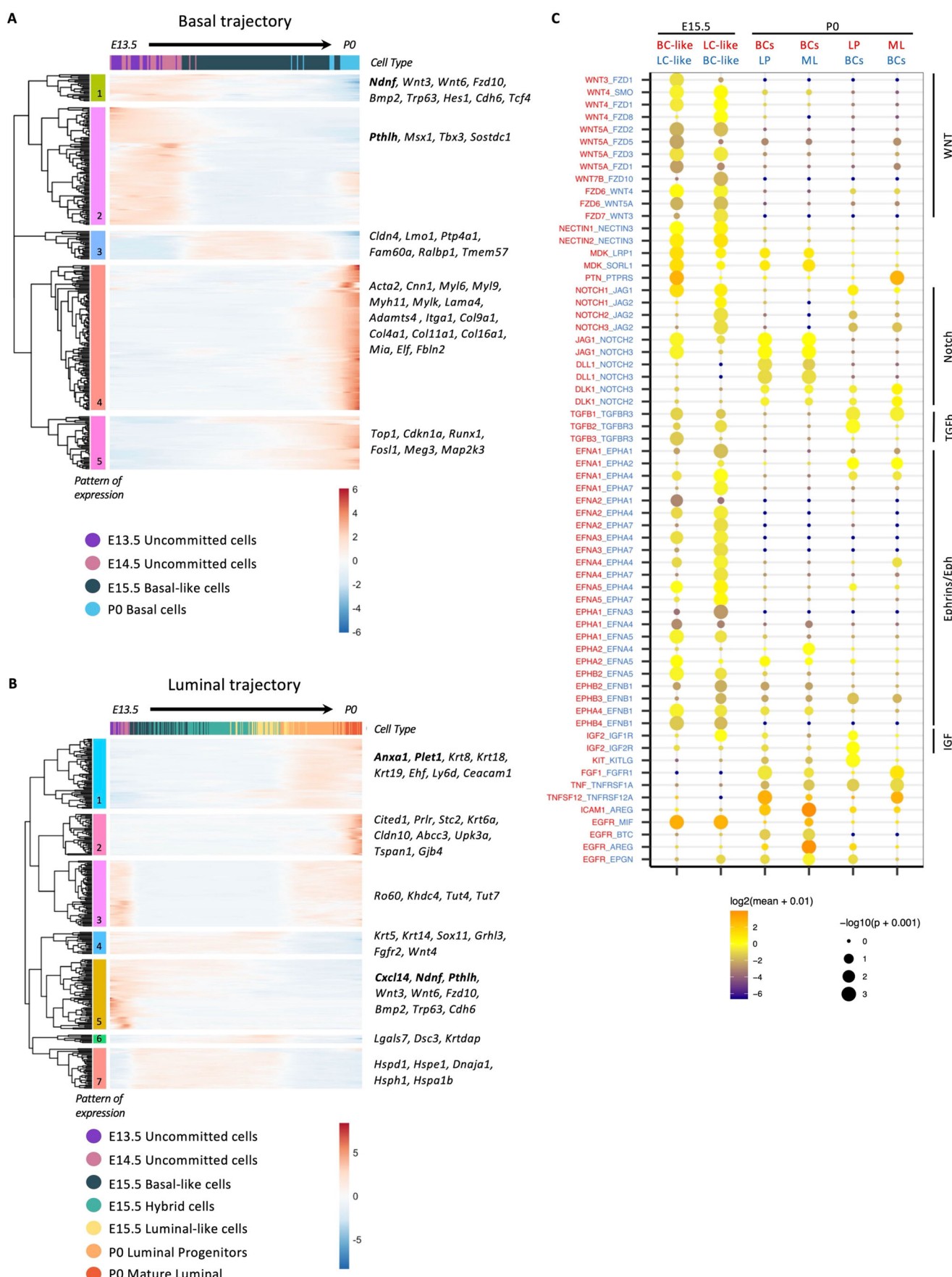

◀ **Figure EV2.   Related to Fig. 1. Pseudotime ordering identifies genes associated with early luminal and basal differentiation and changes in ligand-receptor interaction pairs between basal-like and luminal-like at early and late developmental timepoints.**

(**A**, **B**) Heatmaps illustrating genes exhibiting a differential pattern of expression along the pseudotime (from E13.5 to P0) towards the basal lineage (**A**) or the luminal lineage (**B**). Genes (rows) are clustered based on the dendrogram on the left and color-coded by their expression levels (from blue to red). The gene expression levels were smoothed using the generalized additive model (GAM) and scaled by row. Genes of interest are indicated on the right. Each set of genes with a specific pattern is color-coded on the left: 5 distinct patterns in the basal lineage (**A**) and 7 unique patterns in the luminal lineage (**B**). (**C**) CellPhoneDB analysis showing predicted ligand-receptor interactions between the two epithelial populations at E15.5, basal-like and luminal-like cells (left side of the dotplot), and between the three epithelial populations at P0; BCs, LP and ML cells (right side of the dotplot) (*P* value < 0.01). Permutation test was used for statistical analysis.

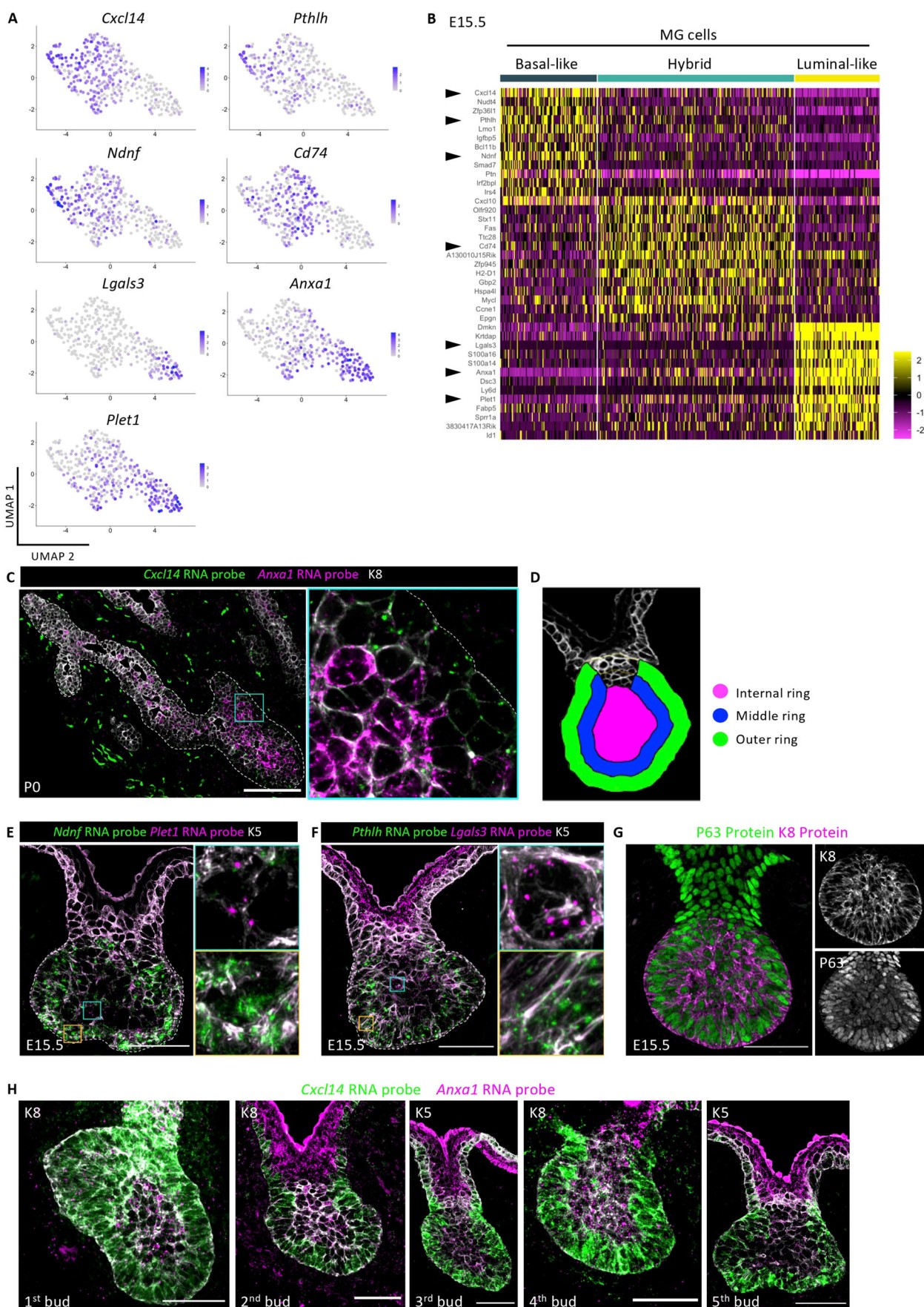

◄ **Figure EV3.   Related to Fig. 2. Identification of novel genes that distinguish lineage-biased embryonic mammary cells.**

(A) UMAP plots showing the expression levels of selected basal (*Cxcl14*, *Pthlh* and *Ndnf*), hybrid (*Cd74*) and luminal (*Lgals3, Anxa1* and *Plet1*) genes at E15.5. (B) Heatmap illustrating the expression of genes specific for each MEC cluster at E15.5. Each column is color-coded according to the cell cluster from Fig. 1B. Black arrowheads indicate genes used in RNAscope experiments. (C) Representative section of a mammary duct at P0 showing the expression of *Cxcl14* (in green) and *Anxa1* (in magenta) detected by RNAscope and of K8 by IF (in white). The white dotted line delineates the BM ($n = 3$). (D) Optical section of a mammary bud at E15.5 illustrating the ROIs: outer ring (in green), middle ring (in blue) and internal ring (in magenta) used for the quantitative analysis. (E, F) Representative sections of embryonic mammary buds at E15.5 showing the expression of *Ndnf* (basal gene, in green) and *Plet1* (luminal gene, in magenta) (E) or *Pthlh* (basal gene, in green) and *Lgals3* (luminal gene, in magenta) (F), detected by RNAscope and immunostained with antibodies anti-K5 (in white) ($n = 2$). (G) Single optical section showing the expression of the luminal epithelial marker K8 (in magenta), and the basal epithelial marker P63 (in green) in an embryonic mammary bud at E15.5. K8 and P63 are co-expressed by all MECs at E15.5. (H) Representative sections of the 5 different embryonic mammary buds (#1 to #5) at E15.5, showing the conserved expression of *Anxa1* (in magenta) and *Cxcl14* (in green) detected by RNAscope. Anti-K5 or anti-K8 immunostaining delineate the mammary bud epithelium (in white) ($n = 2$). Data information: scale bars: 100 µm in (C) and 50 µm in (E–H).

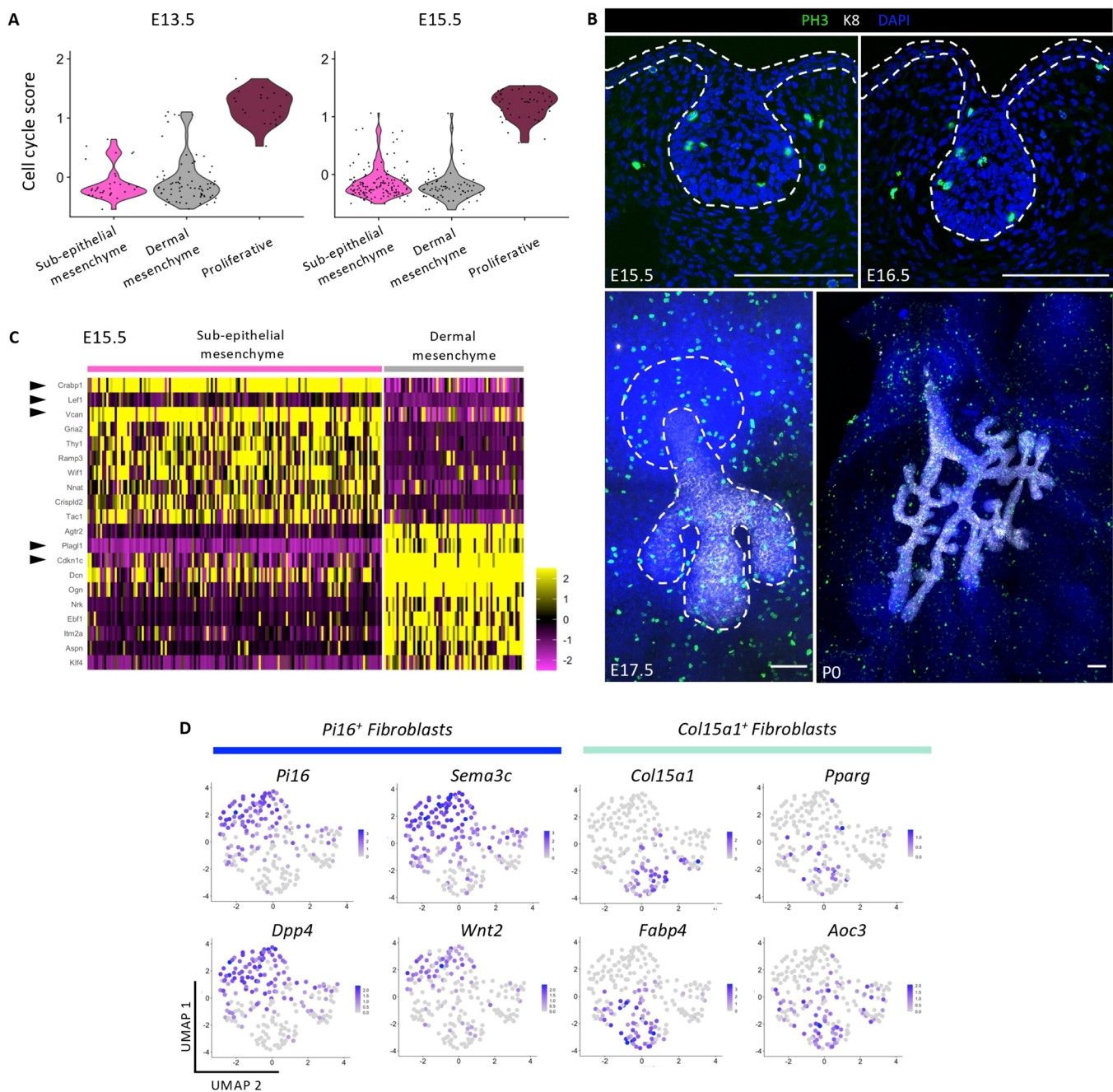

**Figure EV4. Related to Fig. 3. The heterogeneity of mesenchymal cells increases at birth.**

(A) Violin plots representing the cell cycle score in each mammary mesenchymal cluster at E13.5 and E15.5. $n = 129$ cells at E13.5; $n = 252$ cells at E15.5. (B) Representative sections of mammary bud at E15.5 and E16.5 and whole-mount staining at E17.5 and P0 showing PH3+ cells (in green), K8 (in white) and DAPI (in blue) ($n = 2$). Dotted lines delineate the BM (in white). (C) Heatmap illustrating the expression of genes specific for each mesenchymal cluster at E15.5. Each column is color-coded according to the cell cluster from Fig. 3A. The black arrowheads indicate the genes that were further investigated for their specific expression in sub-epithelial or dermal mesenchyme. (D) UMAP plots from Fig. 3A illustrating the expression of cluster-specific genes in mesenchymal cells at P0. Scale bars: 100 μm in (B).

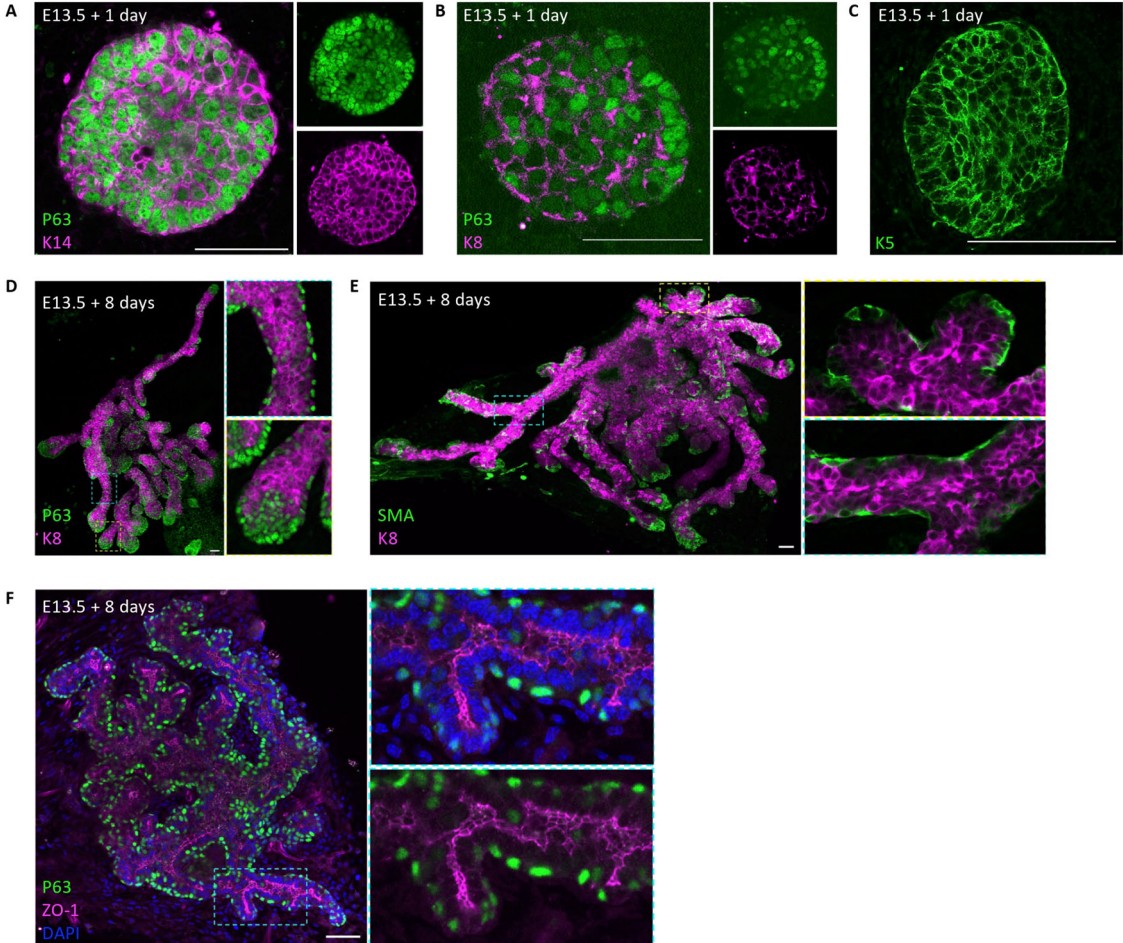

**Figure EV5. Related to Fig. 5. Mammary bud *ex vivo* cultures recapitulate embryonic mammary morphogenesis and epithelial lineage segregation.**

(**A–C**) Representative images of mammary embryonic buds dissected at day E13.5 and cultured ex vivo for 1 day, immunostained for the following lineage markers: P63 (in green) and K14 (in magenta) (**A**), P63 (in green) and K8 (in magenta) (**B**), and K5 (in green) (**C**). (**D–F**) Representative images of mammary embryonic buds dissected at day E13.5 and cultured ex vivo for 8 days, immunostained for the following lineage and polarity markers: P63 (in green) and K8 (in magenta) (**D**), α-SMA (in green) and K8 (in magenta) (**E**), and P63 (in green) and ZO-1 (in magenta) (**F**). Data information: scale bars: 50 µm (in **A–C**), 100 µm (in **D–F**) (*n* = 3).

