## [Peer Review File · The EMBO Journal]

Spatially distinct epithelial and mesenchymal cell subsets along progressive lineage restriction in the branching embryonic mammary gland

Claudia Carabaña, Wenjie Sun, Camila Veludo Ramos, Mathilde Huyghe, Robin Journot, Aurélien Maillot, Meghan Perkins, Fatima Hartani, Marisa Faraldo, Bethan Lloyd-Lewis, and Silvia Fre

Corresponding authors: Silvia Fre (silvia.fre@curie.fr) , Bethan Lloyd-Lewis (bethan.lloyd-lewis@bristol.ac.uk)

Review Timeline:

Submission Date:	25th Jul 23
Editorial Decision:	10th Sep 23
Appeal:	26th Jan 24
Editorial Decision:	22nd Mar 24
Revision Received:	6th Apr 24
Accepted:	17th Apr 24

Editor: Daniel Klimmeck

Transaction Report:

Dear Silvia, dear Bethan,

Thank you for submitting your manuscript for consideration by the EMBO Journal. Please accept again my sincere apologies for the unusual delay with the evaluation of your study at this time of the year. As mentioned, this is due to unusually delayed referee feedback as well as detailed discussion in the editorial team on the case. Your study has been sent to three referees and we have now received reports from all of them, which I copy below. In light of their comments, I am afraid we decided that we cannot offer publication in The EMBO Journal.

As you will see the experts state that the findings will as such be of interest to the field. At the same time, they also express substantial reservations with the analysis which in our view preclude further consideration by the EMBO Journal. Referee #3 points to major issues regarding the advance provided in light of previous reports by your team and others (ref#3, standfirst, pt.5; see also ref#1). This referee also states that the underlying bases for positional fate specification of embryonic mammary gland progenitors and their interplay with mesenchymal cell populations remains too prematurely explored (ref#3, standfirst, pts.1-3; see also ref#1). Further, referee #1 points to limitations in the functional exploration of the three MEC populations (ref#1, standfirst, see also ref#2) and in addition has major concerns on the robustness of the single-cell analysis.

Given these overall negative opinions and considering that we need strong support from the referees to move on, - as you know EMBO Journal relies on offering one major round of revisions until a publishable version of the work is achieved, I am afraid we have concluded cannot offer to publish your study in The EMBO Journal.

If it is in your interest, I am happy to explore suitability of the study for our sister journals. Please let me know if that is the case.

I regret to have to disappoint you at this time. I appreciate your interest and again apologise for the protraction.

Best regards,
Daniel

Daniel Klimmeck, PhD
Senior Editor
The EMBO Journal

Referee #1:

Using a Notch1-creERT2 mouse model to conduct in vivo lineage tracing, this group has previously shown that a subset of Notch1-expressing cells was multipotent between E12.5-13.5, but switched to a luminal-restricted cell fate by E15.5 (Lilja et al., NCB, 2018). In this manuscript, they performed single-cell RNA-seq on both epithelial and stromal cells of mammary glands across three embryonic development times (E13.5, E14.5 and E15.5) and P0. In line with their previous findings, they proposed that MaSCs have already exhibited lineage commitment to basal and luminal lineages at E15.5. They defined the spatial location of committed cells within developing mammary glands at E15.5 while the expression of Notch1 somehow was not identified as a marker for luminal lineage commitment at E15.5 in this manuscript. In contrast to this manuscript, the previous studies only detected a single population around this developmental stage. This is likely because proper integration of all MECs from the different development stages was not conducted for the analysis of the single-cell profiling data in this manuscript. I believe the integration analysis of all data is necessary in order to convincingly define the different 3 clusters of MECs at E15.5. Moreover, there is no functional or lineage tracing analysis on the lineage relationship between the so-called three subsets of MECs in E15.5. The manuscript does not provide insights into the molecular mechanisms how the cell fates are switched at E15.5. They also identified two distinct spatially restricted subsets in the mammary mesenchyme and proposed the mesenchymal FGF10 promotes embryonic branching morphogenesis. However, FGF10 is expressed in dermal mesenchyme, but not the sub-epithelial mesenchyme that is next to so-called basal-like cells at E15.5. It remains unclear how FGF10 travels across the sub-epithelial mesenchyme to affect MECs. The manuscript also did not really address how positional clues dictate cell fate specification as stated in the title.

Referee #2:

In the present manuscript, Carabana et al report single cell analysis of FACS-sorted cells from mouse mammary buds harvested at e 13.5, 14.5, 15.5 and P0. The authors seek to understand when cell lineages are determined based on RNA profiles. The authors corroborate the findings of G. Wahl's group that cell lineages are determined before birth based on scATACseq and

show convincingly that such different populations are present at e15.5 but not e13.5. These results contradict previous findings, in which a different FACS sorting strategy was used which suggested that cell fates were determined at birth. Based on new markers the authors map the different cells to the embryonic buds. The authors also identify different stromal cell subpopulations that change over time.

It is nowadays very difficult to obtain funding for this type of basic research into developmental processes and as such this high quality work albeit being somewhat descriptive is highly commended for publication.

Major points:

1. Some info is missing. The authors mention that mice are in a mixed genetic background; which precisely? After all there are strain difference in mammary gland development. and it would be important that the authors mention the work of J. Veltmaat who has shown that there are differences between different pair of mammary buds during development, i.e. thoracic and inguinal buds develop at different pace and rely differentially on different signalling pathways.

The authors mention Line 91 that mammary buds were pooled, more info is needed on which mammary buds (#1-5) were used and whether all numbers equally represented at different time points. More images of the buds in different locations could be shown in suppl. material for the readers' instruction.

2. The major conclusions would be strengthened if not merely based on the single cell RNA sequencing but validated by an alternative approach.

A: Figure 4A: conclusion that proliferative cell populations in the stroma are not present any more at P1 should be confirmed by Ki67 or PHH3 labeling of mammary glands at the different developmental stages.

B: Figure 4E: conclusion that there are different subpopulations proximal and distal to the bud is based on single marker. Other genes should be used, for lack of antibody this can be looked at mRNA level, RNAscope.

Also where are these cells located in the P1 glands.

What about the P0 specific population? Where are the marker genes expressed?

3. Suppl. Figure 1 shows few cells for e14.5. It seems to this reviewer that it is better to leave it out and as noted elsewhere expand on the analysis of individual genes at the level of histological sections.

4. The authors state that they removed cycling cells from parts of the analysis. This is a bit confusing. Does this apply to just one part of the analysis or throughout? Please explain.

5. In plotting the developmental cell changes and interpreting them it would be useful to know more about the in vivo dynamics of these processes. What is known about cell proliferation and cell death at the different time points? If there is no literature to refer to, some IHC to address this would be a useful addition and make this ms more of a reference point.

6. The nice final model omits P0. A second panel showing the populations dynamics in the stroma would be useful.

Minor:

1. Lines 48f :

"an inner layer of polarized luminal cells (LCs), facing the ductal lumen, that encompass hormone receptor (namely Estrogen (ER) and Progesterone (PR) receptors) expressing and non-expressing subpopulations."

Statement is too global and needs to be corrected. Initially there is no HR expression, then the receptors are fairly ubiquitously expressed in the luminal cells; the pattern of positive and negative cells appears only with puberty.

"How changes in mammary tissue architecture translate into differential gene expression patterns that drive the lineage specification of individual cells during development remains unknown in many tissue contexts."

Should read:

that are characteristic of different lineages

Line 114 ff: indicate in the text which genes were identified that had not previously been shown.

Line 137: provide a link to the webtool here

Lines 147 ff Based on what parameters do the authors conclude that there is "vast cellular heterogeneity"?

Referee #3:

Peer-reviewing EMBO Journal
#EMBOJ-2023-115114

Carabaña and colleagues investigate the fate of embryonic mammary epithelial stem cells across development, and how morphogenesis and timing contribute to the shift from bipotency to unipotency. The authors profiled the mouse embryonic mammary gland by scRNA-Seq at 4 different developmental stages E13.5; E14.5; E15.5; and Birth, and identified E15.5 as the key stage where MaSC lose their bipotent hybrid identity and become unipotent, coinciding with the initiation of MG branching. Spatial analyses revealed newly formed embryonic luminal like and basal-like cells are located in mutually exclusive regions of the bud, with basal cells in close proximity to the stromal compartment. Finally, they identified stromal derived FGF10 as a key factor inducing branching.

The article is well written, the methods section is detailed, the figures and legends are well presented, which makes altogether a manuscript easy to follow. However, some of the findings from this work are not novel (see below) in regard to the existing literature, and some findings deserve a more in-depth analysis, before being suitable for a publication in a journal like EMBO-J which aims at a broad readership. Here are my comments below:

Major concerns:

Since the back-to-back publication in NCB by Wuidart et al., Lilja et al., (Fre Lab) 2018, we know that Embryonic MaSC express a luminal basal hybrid signature inferred from scRNA-Seq profiling (Wuidart) and that lineage segregation happens at the embryonic E15 stage, which coincides with the initiation of branching in the MG. This present article elegantly brings refined timing of the development stages during which the bipotency to unipotency switch occurs, however the transcriptional analysis, despite unveiling additional markers during this transition, brings limited information about the molecular mechanisms driving lineage segregation, and is rather descriptive.

The most interesting aspects of this work is the characterization of the dermal/sub epithelial mesenchymes and the analysis of the receptor/ligand crosstalk between the two types of mesenchyme and the mammary epithelial cells, at the time of lineage segregation. Given the spatial localization of the basal-like cells within the bud, it is likely that these cells are exposed to the highest concentration of extra cellular proteins and ligands (such as FGF10) arising from the stromal region and therefore likely to respond and differentiate along the basal lineage, while the luminal cells located in the center of the bud receive limited amount of stromal ligands and therefore are less likely to interact with the stromal compartment. Their differentiation towards the luminal lineage could therefore passively result from the lack of stromal input. This is probably what lead the authors to present only the receptor/ligand interactome between the basal cells and the stromal compartment (figure S4).

1) It is possible that during the switch from bipotency to unipotency, basal cells (in response to stromal stimulation) can secrete factors that orient the fate of luminal cells. A similar comparative luminal/basal CellPhoneDB analysis would be needed between E14.5, before the switch occurs and at E15.5 during the potency switch, to answer this question.

2) It is known that cells can differentially respond to ligands depending on their concentration. This is typically described in vascular biology where low concentrations of semaphorins or VEGF trigger different biological response than high concentrations. Therefore, it would be worth to analyze the luminal-like versus stromal interactions using CellPhoneDB and see whether some specific ligand receptor pair would pop up in the analysis.

3) How is the ligand/receptor interaction landscape changing between basal cells and the mesenchyme at E14.5, before the switch occurs as compared to E15.5 during the potency switch? It is worth comparing the two stages as it would narrow down the factors involved in the epithelial / stromal crosstalk. Also, I do not know to what extend the limited number of cells profiled by scRNA-Seq limits the power of this study, and also how other tools such as NicheNet (<https://www.nature.com/articles/s41592-019-0667-5>) would perform with such dataset.

4) Additionally, the authors could go back to their scRNA-Seq dataset, extract ligand encoding genes expressed at high level in both sub epithelial and dermal mesenchyme compartments, and functionally annotate them. This would reinforce the CellphoneDB analysis and help narrowing down the stromal factors that actively contribute to fate decision and branching. For example, is FGF10 expressed at all in the mesenchyme (check the scRNA-Seq dataset)? And is FGFR2 expressed in the basal compartment? In the luminal compartment? Also, multiple Wnt/Fzd pairs also pop up in the CellphoneDB analysis, but what Wnt ligands (if any) are actually expressed by the mesenchyme?

5) The authors investigate how FGF10/FGFR2 crosstalk drives branching using a very powerful ex vivo model of explant culture. However, the contribution of FGF10 to branching in the mammary gland already received several experimental support, such as : <https://www.ncbi.nlm.nih.gov/pmc/articles/PMC4199126/>, <https://pubmed.ncbi.nlm.nih.gov/9428423/>, <https://www.frontiersin.org/articles/10.3389/fcell.2020.00415/full>, <https://journals.biologists.com/dev/article/129/1/53/17506/Role->

This is limiting the novelty of the study in this regard, and the validation of the whole scRNA-Seq profiling. Also, in their model, FGF10 ligands from the stroma of the explant might also be driving branching and can mask the biological effects of exogenously added FGF10. Hence, wouldn't it be more suitable to use a FGFR inhibitor (at a sub lethal dose) (or FGF10 blocking ab?) and assess how branching is affected ex vivo? How cell fate is affected (with for example absence of lineage switch and/or presence of cell bearing markers of both lineage)? Indeed, the fact that FGF10 signalling contributes to branching does not necessarily mean it drives the switch from bipotency to unipotency. The reasons why the potency switch and the branching coincide remain yet to be elucidated. Adding FGF10 ex vivo exposes the luminal like and basal like cells to the same concentration of FGF10, meaning their position in regards to the stroma (and therefore the concentration of FGF10 to which they are exposed to in vivo) does not matter anymore, can the authors comment on that ? Finally, given that the number of branches in explant does not change upon FGF10 stimulation (only their diameter), the title of figure 5 appears to be inaccurate.

The title of this work emphasizes that "positional cues underlie cell fate specification", and given the results, it seems it's the position in regard to the mesenchyme that dictates lineage segregation. Therefore, the authors should strengthen this part of the work.

Minor concerns:

1) Regarding the gating strategy (Fig Sup 1), how can the authors be sure that all the embryonic mammary epithelial cells are strictly Epcam high? What about Epcam mid, or low as found in normal adult mammary gland? Can the author comment on the gating strategy?

2) The experimental data from the literature supporting the bi to unipotent progenitor switch post E15.5, including this article, is very convincing. However, I recommend that the discussion of this article is tempered by at least two key points : 1) the existence of bipotent adult MaSC have received several experimental support from different labs and have been associated with different markers such as Dll1 (<https://www.science.org/doi/10.1126/science.aan4153>), PROCR (<https://pubmed.ncbi.nlm.nih.gov/25327250/>), or Krt5 (<https://www.nature.com/articles/nature12948>). The controversy regarding their existence and activity in the adult MG therefore remains a topic of vivid discussion, and it would be adequate to at least place this work in the broader context. 2) how do these findings relate to the human breast biology where MaSC are described in the adult gland, and where cells co-expressing luminal and basal markers have been clearly defined (notably by the work of Tan Ince and others)? How is this bi to unipotency switch relevant to the human biology?

3) Figure 2 as it is, provides limited information and could be put in supplementary figures, while Figure S4 deserved to be among the main figures, together with more in depth analysis of the ligand/receptor interactions. Figure 6 could also be merged with Figure 5.

** As a service to authors, EMBO Press provides authors with the possibility to transfer a manuscript that one journal cannot offer to publish to another EMBO publication or the open access journal Life Science Alliance launched in partnership between EMBO Press, Rockefeller University Press and Cold Spring Harbor Laboratory Press. The full manuscript and if applicable, reviewers' reports, are automatically sent to the receiving journal to allow for fast handling and a prompt decision on your manuscript. For more details of this service, and to transfer your manuscript please click on Link Not Available. **

EMBOJ-2023-115114; Response to Referee reports

Original comments in black, authors' replies in blue.
Changes are highlighted in blue in the revised manuscript.

Referee #1:

Using a Notch1-creERT2 mouse model to conduct in vivo lineage tracing, this group has previously shown that a subset of Notch1-expressing cells was multipotent between E12.5-13.5 but switched to a luminal-restricted cell fate by E15.5 (Lilja et al., NCB, 2018). In this manuscript, they performed single-cell RNA-seq on both epithelial and stromal cells of mammary glands across three embryonic development times (E13.5, E14.5 and E15.5) and P0. In line with their previous findings, they proposed that MaSCs have already exhibited lineage commitment to basal and luminal lineages at E15.5. They defined the spatial location of committed cells within developing mammary glands at E15.5 while the expression of Notch1 somehow was not identified as a marker for luminal lineage commitment at E15.5 in this manuscript.

We thank the reviewer for nicely summarizing our study. In response to the last point on the expression of Notch1, we would like to clarify that in our previous work (Lilja et al., NCB, 2018) Notch1 was used to lineage trace cells (using our Notch1-CreERT2 mice): at E15.5, Notch1 expression labeled unipotent clones that were equally either basal or luminal, indicating that the expression of the Notch1 receptor is not yet segregated to luminal cells at this embryonic stage. Indeed, by RNA FISH we have shown that Notch1 is homogeneously expressed in all cells of the mammary bud at E15.5 (Lilja et al., NCB, 2018, Suppl. Fig1B – see also Fig. R1A). The lack of segregation in Notch1 expression is also apparent by its comparable levels in all 3 epithelial clusters in our scRNAseq data at E15.5 (see Fig. R1B-C). Therefore, we would not expect to identify Notch1 as a specific marker for luminal lineage commitment at this stage.

Figure for reviewers removed

In contrast to this manuscript, the previous studies only detected a single population around this developmental stage. This is likely because proper integration of all MECs from the different development stages was not conducted for the analysis of the single-cell profiling data in this manuscript. I believe the integration analysis of all data is necessary in order to convincingly define the different 3 clusters of MECs at E15.5.

We concur with the Referee with the necessity of data integration; accordingly, in the originally submitted manuscript we had indeed integrated our data from each of the four developmental timepoints. We presented the results of this integration in 3D in Fig. 1F-G. Moreover, integrating all data points enabled us to delineate the differentiation trajectories shown in Fig. 1G using a minimum spanning tree (MST) connecting each cell cluster. We apologize that this was unclear from the text, which we have now made more explicit in the methods (page 30) and legend of Figure 1. Re-integrating our data from all 4 time points in 2D into a single UMAP plot also provides convincing evidence for the existence of 3 different MECs clusters at E15.5 (Fig. R2), which we can include as an Extended view Figure if preferred by the Referee.

Figure for reviewers removed

To further address the Referee's concerns, we have performed a comparative analysis with previous studies, which confirmed our findings on lack of MECs commitment to the basal or luminal lineage at E13.5-14.5 and on the presence of distinct lineage-biased clusters from E15.5, as further elaborated below.

First, we re-analysed the datasets by Wuidart et al. 2018 with our bioinformatic pipeline and applied the same basal and luminal score used for our data, which confirmed the presence of a single homogeneous cell cluster at E14.5 (Fig. R3A), comparable to the single cluster we identified at the same developmental time (see Expanded view Figure 1B). Also corroborating our results at E13.5 and E14.5, when we applied the basal and luminal score, we observed that this unique E14 cluster contains cells mostly co-expressing basal-like and luminal-like genes, consistent with the 'hybrid' lineage transcriptional signature identified by Wuidart and co-authors (Fig. R3A).

Secondly, when we employed our analysis pipeline to analyse the E16 and E18 dataset produced by Girardi et al. 2018, three transcriptionally distinct cell subsets could be distinguished (Fig. R3B-C), also confirming our results at E15. Finally, re-clustering analysis of the MECs derived from Pal et al. 2021, identified again three transcriptionally distinct cell populations at E18 (Fig. R3D). Collectively, our detailed comparative analysis illustrates a lack of discrepancy between our results and previous studies. Also supporting our findings, recent snATAC-seq analyses (Chung et al. 2019) revealed that embryonic mammary cells at E18 exhibit either a basal-like or luminal-like chromatin accessibility profile, features that likewise indicate partial lineage specification prior to birth, strongly supporting our findings.

It is important to note however that methods of "forced" integration can artificially create similarities in otherwise distinct cell clusters, which may explain (along with differences in FACS sorting strategies as noted by Referee 2) why the work from Girardi et al and from Pal et al identified a single population after E15. It's imperative therefore that the presence of distinct cell populations identified by scRNAseq analyses are carefully validated experimentally using alternative approaches, such as by smRNA-FISH as performed in the present study.

We discuss these points extensively in the text (lines 425-440).

Figure for reviewers removed

Moreover, there is no functional or lineage tracing analysis on the lineage relationship between the so-called three subsets of MECs in E15.5. The manuscript does not provide insights into the molecular mechanisms how the cell fates are switched at E15.5.

We completely agree with the Referee on the first point; lineage tracing and functional studies using the new molecular markers revealed in our study will be fantastic to unequivocally establish the potency and function of the early segregated progenitors that we identified at E15.5. We believe, however, that the scRNAseq atlas and cell-cell interaction analysis produced in this work (as detailed in our response to Referee 3) provides new insights into the molecular mechanisms and signals important for embryonic cell fate specification and/or branching morphogenesis, which we and others can advance on in future studies.

The discovery of novel early lineage markers in the present study also represents an advance in the field, opening up the possibility of establishing novel genetic tools to explore cell fate specification not only in the mammary epithelium, but also in other glandular epithelia sharing a similar stem cell hierarchy. We believe it imperative therefore to publish our results in a timely manner. We are currently developing two new CreERT2 mouse lines that will enable us to trace the fate and potency of early mammary luminal and basal progenitors; however, the protracted time necessary for generating and validating these new genetically engineered mouse models does not allow us to exhaustively address this point within the timeframe of this revision.

They also identified two distinct spatially restricted subsets in the mammary mesenchyme and proposed the mesenchymal FGF10 promotes embryonic branching morphogenesis. However, FGF10 is expressed in dermal mesenchyme, but not the sub-epithelial mesenchyme that is next to so-called basal-like cells at E15.5. It remains unclear how FGF10 travels across the sub-epithelial mesenchyme to affect MECs.

FGF10 is a soluble protein that can travel long distances by diffusion; indeed there are numerous examples in the literature of long-range signaling and gradient formation mediated by FGF ligands (see for a review i.e. “Mechanisms of FGF gradient formation during morphogenesis”, doi:10.1016/j.semcd.2015.10.004). It is therefore not surprising that FGF10 produced by the dermal mesenchyme - which, based on our smRNA-FISH experiments, starts at a distance of only 12-20 μm from the epithelial bud - can affect MECs.

The manuscript also did not really address how positional clues dictate cell fate specification as stated in the title.

While we believe that our study puts solid premises for addressing how positional cues within the epithelial bud and factors from the surrounding environment can direct mammary epithelial cell fate specification, we appreciate the Referee’s point and have modified our title accordingly to: “Single cell analysis reveals spatially distinct epithelial and mesenchymal cell subsets in the branching embryonic mammary gland”.

Referee #2:

In the present manuscript, Carabana et al report single cell analysis of FACS-sorted cells from mouse mammary buds harvested at e 13.5, 14.5, 15.5 and P0. The authors seek to understand when cell lineages are determined based on RNA profiles. The authors corroborate the findings of G. Wahl's group that cell lineages are determined before birth based on scATACseq and show convincingly that such different populations are present at e15.5 but not e13.5. These results contradict previous findings, in which a different FACS sorting strategy was used and which suggested that cell fates were determined at birth. Based

on new markers the authors map the different cells to the embryonic buds. The authors also identify different stromal cell subpopulations that change over time.

It is nowadays very difficult to obtain funding for this type of basic research into developmental processes and as such this **high quality work albeit being somewhat descriptive is highly commended for publication.**

We thank the referee for appreciating the quality of our work. With regards to how our results relate to previous findings, please see our response to Referee #1 (pages 1-3) where we have undertaken a detailed comparative analysis of 3 published datasets (Wuidart et al 2018; Giraddi et al 2018 and Pal et al 2021) using our bioinformatic pipeline. This analysis identified distinct cell populations also in the other datasets, at E16 and E18 in the Giraddi and Pal datasets respectively (but not at E14 in the Wuidart data), which corroborates, rather than contradicts our results. The discrepancy may be due to differences in FACS sorting strategies as suggested by the reviewer, or to differences in data integration and analysis approaches, as we discuss in the manuscript (lines 425-440). This point also underscores the importance of carefully validating the existence of distinct cell populations revealed by scRNAseq using *in situ* approaches, such as IHC or smRNA-FISH, which we have accomplished in the present work.

Major points:

1. Some info is missing. The authors mention that mice are in a mixed genetic background; which precisely? After all there are strain difference in mammary gland development. and it would be important that the authors mention the work of J. Veltmaat who has shown that there are differences between different pair of mammary buds during development, i.e. thoracic and inguinal buds develop at different pace and rely differentially on different signalling pathways.

The authors mention Line 91 that mammary buds were pooled, more info is needed on which mammary buds (#1-5) were used and whether all numbers equally represented at different time points. More images of the buds in different locations could be shown in suppl. material for the readers' instruction.

We apologize for the confusion: in the section “Mouse models” in Materials and Methods, we explicitly said that we have used mice of mixed background only for explant cultures and not for scRNAseq analysis, which was performed on WT C57 B6/N animals. For the mixed genetic background mice, they were initially C57 B6/J, then they were crossed to CD1 and the F1 obtained was then backcrossed once on 129 background and then intercrossed currently for 15 generations. We have now included the detailed information on genetic background in the Material and Methods section “Mice” (page 21).

To our knowledge, no study has reported that mouse genetic background affects the timing of mammary embryonic bud development. As highlighted by the Referee, asynchronous development of distinct mammary bud pairs within the same mouse is well-established (and clearly visible at P0 in Figure R4), mainly thanks to the work from the Bellusci and Veltmaat labs (Mailleux et al., 2022; Veltmaat et al., 2004), which we refer to on lines 47-49: “Mouse mammary gland development begins at embryonic day (E) 10 with the formation of bilateral milk lines, followed by the asynchronous appearance of five pairs of epithelial placodes positioned symmetrically at each side of the embryo (Veltmaat et al 2004; Mailleux et al 2002).”

Following the Referee’s suggestion, we now include images of the 5 mammary buds in different locations (Expanded view Fig. EV4H) in the revised manuscript. Importantly, these images show RNAscope analysis for the expression of the two early fate markers we identify, *Anxa1* and *Cxcl14*, at E15, which showed no

overt differences in cell fate segregation among different buds. This provides confidence that this asynchronous development is not a major factor influencing the timing of fate segregation.

We have also now clarified in the Material and Methods section (Lines 657-658) that all 5 mammary bud pairs in each embryo were pooled for scRNAseq, where they were equally represented at each different time point. This was necessary because of the low number of cells that can be recovered upon bud microdissection and cell sorting.

Figure for reviewers removed

2. The major conclusions would be strengthened if not merely based on the single cell RNA sequencing but validated by an alternative approach.

A: Figure 4A: conclusion that proliferative cell populations in the stroma are not present any more at P1 should be confirmed by Ki67 or PHH3 labeling of mammary glands at the different developmental stages.

We apologize for the misunderstanding: in our scRNAseq analysis we did observe a population of proliferative stromal cells also at P0, however it was not scored sufficiently high from the clustering algorithm to produce a distinguishable and independent UMAP cluster. We noticed that the mammary mesenchymal compartment becomes increasingly more heterogenous at birth, and that differences in gene expression between cells appear more pronounced at P0 compared to embryonic days. If this bias in transcriptional programs is further exacerbated, some transcripts of cell cycle genes might not even be detected, and cells would not be scored as cycling cells. Thus, it is plausible that differential expression of proliferative genes may not have met the requirements to generate an independent UMAP cluster representing proliferating cells at P0. In line with the Referee's suggestion, we investigated this point further by immunostaining mammary buds and the surrounding mesenchyme for phospho-histone H3 (a cell proliferation marker) at different developmental stages (namely E15.5, E16.5, E17.5 and P0), which revealed the presence of proliferating stromal cells at P0. We include this data in Expanded view Fig. EV5B.

B: Figure 4E: conclusion that there are different subpopulations proximal and distal to the bud is based on single marker. Other genes should be used, for lack of antibody this can be looked at mRNA level, RNAscope.

We agree and thank the Referee for this valuable suggestion. To address this important point, we have now analyzed the expression of additional markers for the two stromal cell subpopulations (indicated by arrowheads in Expanded view Fig. EV5C), and we have included immunostaining against LEF1 (Figure 3C) and smRNA FISH for Vcan and Crabp1 (Figure 3E), specifically enriched in the sub-epithelial cluster, whereas Cdkn1c expression (Figure 3E) demarcates the dermal cluster, like PLAGL1 (Figure 3D).

Also where are these cells located in the P1 glands.

What about the P0 specific population? Where are the marker genes expressed?

By analyzing the expression of selected E15.5 stromal markers (Vcan, Crabp1 and Cdkn1a) at P0 we could not detect segregation of stromal cells expressing these genes relative to the branched mammary tree. We then tried to spatially distinguish the P0 stromal clusters using specific RNAscope probes, which also failed to reveal spatial segregation of stromal cells. Our scRNAseq data showed that stromal cell composition at P0 is markedly different to that at E15, indicating that mesenchymal cells change significantly during this developmental window. It's therefore not entirely surprising that the limited number of markers selected for analysis failed to discriminate distinct regions at P0. This is further complicated by the fact that the epithelial tree has branched and elongated considerably by this time, making the spatial distribution of cells difficult to define (particularly in 2D tissue sections where we cannot observe the position of branches running below/above ducts included in each section). Given these uncertainties, we prefer not to include the RNAscope data for our selected markers in the P0 stroma.

3. Suppl. Figure 1 shows few cells for e14.5. It seems to this reviewer that it is better to leave it out and as noted elsewhere expand on the analysis of individual genes at the level of histological sections.

While fewer cells were analyzed at E14.5, sequenced reads passed the high-quality threshold set. Importantly, the results we obtained at E14.5 are consistent with findings of a homogeneous population of hybrid embryonic multipotent progenitors (EMP) identified at the same developmental stage by Wuidart et al. Nat. Cell Biol. 2018. Thus, we believe it important to maintain the E14.5 data in the manuscript. For readers to take this into account when interrogating our raw or processed datasets (on GEO and the online open-access tool that we have generated), we included the precise cell numbers for each timepoint in the Methods section (lines 742-743).

4. The authors state that they removed cycling cells from parts of the analysis. This is a bit confusing. Does this apply to just one part of the analysis or throughout? Please explain.

We apologize for the confusion. We have linearly regressed the cycling cell cluster only at E15.5 to remove unwanted variation coming from the cell cycle stage and analyze in more depth the three different epithelial clusters that we identified at this time. Proliferating cells were uniformly distributed among the three E15.5 MEC clusters. We have clarified this point in the revised manuscript (lines 113-115).

5. In plotting the developmental cell changes and interpreting them it would be useful to know more about the in vivo dynamics of these processes. What is known about cell proliferation and cell death at the different time points? If there is no literature to refer to, some IHC to address this would be a useful addition and make this ms more of a reference point.

We thank the Referee for this suggestion and for proposing that our manuscript become a reference point for the characterization of mammary embryonic development. In the revised manuscript, we now provide immunofluorescence data for proliferative cells (PH3) in both epithelium and stroma at different developmental times (E15.5, E16.5, E17.5 and P0) in Expanded view Figure EV5B. We also cite two relevant recently published studies on cell proliferation in the early mammary embryonic bud from the group of Dr M. Mikkola (Myllymaki et al. JCB 2023; <https://doi.org/10.1083/jcb.202209005>; Lan et al. [bioRxiv preprint doi.org/10.1101/2023.04.24.538064](https://doi.org/10.1101/2023.04.24.538064)).

Regarding apoptosis, we have also performed immunofluorescence staining for cleaved caspase-3 and observed very few and sporadic apoptotic cells in the female mammary bud at E15.5 and P0 (Figure R5). This is consistent with a published study showing that female mammary buds at E14.5 are negative for TUNEL staining, while male buds are positive (indicative of male mammary regression: doi: 10.1016/j.ydbio.2006.03.030). Based on these studies, we believe that cell death does not significantly contribute to the earliest stages of mammary embryonic morphogenesis.

Figure for reviewers removed

6. The nice final model omits P0. A second panel showing the populations dynamics in the stroma would be useful.

We thank the Referee for appreciating our model and for their suggestions. Accordingly, we have included the P0 developmental time point for the epithelium in our final model in revised Figure 6B. However, as the population dynamics in the stroma are less linearly and hierarchically organized (as discussed above) we prefer not to draw a scheme for the stromal cells, as we feel it would be too hypothetical at this stage.

Minor:

1.Lines 48f :

"an inner layer of polarized luminal cells (LCs), facing the ductal lumen, that encompass hormone receptor (namely Estrogen (ER) and Progesterone (PR) receptors) expressing and non-expressing subpopulations." Statement is too global and needs to be corrected. Initially there is no HR expression, then the receptors are fairly ubiquitously expressed in the luminal cells; the pattern of positive and negative cells appears only with puberty.

We have corrected the text to clarify that the two luminal mature and progenitor populations arise at puberty (lines 56-58).

"How changes in mammary tissue architecture translate into differential gene expression patterns that drive the lineage specification of individual cells during development remains unknown in many tissue contexts."

Should read:

that are characteristic of different lineages

Noted, the change is included at line 97.

Line 114 ff: indicate in the text which genes were identified that had not previously been shown.

We apologize for the confusion. In the text we wrote: “This analysis also identified genes that had not been previously ascribed to distinct mammary BC or LC populations.” Accordingly, the answer to this referee’s point is: all genes we identified **as being characteristic of E15 basal-like and luminal-like cells** had not been previously shown. In fact, our analysis offers the capacity of distinguishing lineage-biased cells as early as at E15 by their differential expression of a whole set of genes, some of which may be novel, but others were simply not ascribed to a specific lineage in the embryonic mammary epithelium.

To clarify this point, we have now included the following paragraph in the revised text (lines 127-134):
“As expected, lineage markers commonly used to distinguish LCs (Krt8, Krt18) from BCs (Krt5, Trp63) in the postnatal mammary gland were co-expressed in all 3 MECs clusters at E15.5 (Figure EV1F). Other genes with reported lineage-specific expression (Kendrick et al. 2008) but no well-established role in adult LCs (Anxa1, Ly6d) and BCs (Lmo1, Pthlh, Cxcl14) were found to be already differentially expressed at E15.5, identifying them as potential early lineage markers. Importantly, the basal-like cluster, but not luminal-like cells, was particularly enriched in genes encoding for factors known to be essential for mammary embryonic morphogenesis (Pthlh, Wnt10b, Sostdc1, Bmp2, Msx1). Furthermore, novel genes with no reported expression in MECs were also found specifically in the basal-like cluster (Ptp4a1, Fam60a, Ralbp1) or in the luminal-like cluster (Grhl3, Krt18).”

Line 137: provide a link to the webtool here

We now provide in the Methods section “Data and software availability” (lines 800-804) the link to our interactive online tool to explore our dataset and evaluate the dynamic expression of any gene of interest along the developmental atlas that we have generated: https://sunwjie.shinyapps.io/Embryo_scRNASeq/
In line with our commitment to data sharing, this link will be available publicly and open access to ensure that our data can be thoroughly mined by interested researchers upon acceptance of this manuscript. We have also deposited all raw data files in GEO, in an attempt to reduce the need for the sequencing experiments performed in our work to be duplicated by other researchers.

Lines 147 ff Based on what parameters do the authors conclude that there is "vast cellular heterogeneity"?

The “vast cellular heterogeneity” referred to our observations of three different cell clusters by UMAP analysis at E15.5, a developmental time when embryonic mammary cells were assumed to be homogeneous based on the uniform expression of all previously known lineage markers at this stage. We have deleted “vast” in the revised text (line 167).

Referee #3:

Carabaña and colleagues investigate the fate of embryonic mammary epithelial stem cells across development, and how morphogenesis and timing contribute to the shift from bipotency to unipotency. The authors profiled the mouse embryonic mammary gland by scRNA-Seq at 4 different developmental stages E13.5; E14.5; E15.5; and Birth, and identified E15.5 as the key stage where MaSC lose their bipotent hybrid identity and become unipotent, coinciding with the initiation of MG branching. Spatial analyses revealed newly formed embryonic luminal like and basal-like cells are located in mutually exclusive regions of the bud, with basal cells in close proximity to the stromal compartment. Finally, they identified stromal derived FGF10 as a key factor inducing branching.

The article is **well written**, the **methods section is detailed**, the **figures and legends are well presented**,

which makes altogether a **manuscript easy to follow**. However, some of the findings from this work are not novel (see below) in regard to the existing literature, and some findings deserve a more in-depth analysis, before being suitable for a publication in a journal like EMBO-J which aims at a broad readership. Here are my comments below:

We thank the Referee for their positive feedback on our work. His/her insightful comments have helped us substantially strengthen the impact, novelty and conclusions of our study. A detailed point-by-point response to all comments is provided below.

Major concerns:

Since the back-to-back publication in NCB by Wuidart et al., Lilja et al., (Fre Lab) 2018, we know that Embryonic MaSC express a luminal basal hybrid signature inferred from scRNA-Seq profiling (Wuidart) and that lineage segregation happens at the embryonic E15 stage, which coincides with the initiation of branching in the MG. This present article elegantly brings refined timing of the development stages during which the bipotency to unipotency switch occurs, however the transcriptional analysis, despite unveiling additional markers during this transition, brings limited information about the molecular mechanisms driving lineage segregation, and is rather descriptive.

We thank the Referee for their appreciation of our results, and the fact that we have conclusively identified the timing of development at which lineage segregation occurs in this tissue. It is important to note that while our (Lilja et al 2018) and Wuidart et al 2018 previous lineage tracing studies and theoretical analysis implied that the switch from multipotency to unipotency of MaSCs occurs during embryonic development at around E15.5, no molecular evidence supporting the existence of early committed embryonic cell populations at this timepoint, nor of their specific location within the mammary bud, was available. In this regard, findings from the present study are novel and advance the current state-of-the-art in the field.

The most interesting aspects of this work is the characterization of the dermal/sub epithelial mesenchymes and the analysis of the receptor/ligand crosstalk between the two types of mesenchyme and the mammary epithelial cells, at the time of lineage segregation. Given the spatial localization of the basal-like cells within the bud, it is likely that these cells are exposed to the highest concentration of extra cellular proteins and ligands (such as FGF10) arising from the stromal region and therefore likely to respond and differentiate along the basal lineage, while the luminal cells located in the center of the bud receive limited amount of stromal ligands and therefore are less likely to interact with the stromal compartment. Their differentiation towards the luminal lineage could therefore passively result from the lack of stromal input. This is probably what lead the authors to present only the receptor/ligand interactome between the basal cells and the stromal compartment (figure S4).

1) It is possible that during the switch from bipotency to unipotency, basal cells (in response to stromal stimulation) can secrete factors that orient the fate of luminal cells. A similar comparative luminal/basal CellPhoneDB analysis would be needed between E14.5, before the switch occurs and at E15.5 during the potency switch, to answer this question.

We thank the Referee for raising this interesting point. As requested, we have now performed CellPhoneDB analysis of interaction pairs between luminal-like and basal-like epithelial cells at E15.5 (Expanded view Figure EV3). We cannot, however, test the luminal/basal communication before the switch occurs, at E13 or E14, since luminal and basal-like populations cannot be discerned at these early stages, when the

epithelial cells still comprise a rather uniform UMAP cluster. To further extend our analysis, we have now also analyzed the interaction between more committed luminal (LP and ML cells) and basal cells at P0 (Expanded view Figure EV3). This additional comparative analysis indicates that one of the major pathways governing this crosstalk is Notch signaling, with the Notch2 and Notch3 receptors specifically expressed in luminal-like cells and Notch ligands (i.e. Jagged2) being enriched in basal-like cells already at E15 (Expanded view Figure EV3). This finding is supportive of work from our lab and others, implicating Notch signaling as an essential pathway for dictating the binary decisions between luminal and basal cell fates. The new CellPhoneDB analysis also highlighted the communication between different Eph receptors expressed by basal-like cells (i.e. EPHA7) and several Ephrins (such as EFNA1) enriched in luminal-like cells at E15. This is interesting since Ephrins/Eph signaling plays an important role in cell guidance during embryonic development (Perez and Getsios, 2014). Finally, our results show that numerous components of the Wnt pathway are expressed in the E15.5 mammary bud. It has been previously shown that Wnt10b expression is one of the earliest markers localized in the mammary epithelium (Chu et al., 2004). Corroborating these findings, our analysis identified that Wnt3 (Expanded view Figure EV3) and Wnt10b (Figure 4B) are enriched in BC-like and hybrid cells (but not in LC-like cells). Interestingly, at P0, some of these embryonic inter-epithelial interactions are predicted to be strongly attenuated (Wnts, Ephrins/Eph) while other pathways known to operate in the adult gland clearly emerge (AREG, Kit, TNF). We have included these new results in the revised manuscript (Expanded view Figure EV3, lines 201-218).

2) It is known that cells can differentially respond to ligands depending on their concentration. This is typically described in vascular biology where low concentrations of semaphorins or VEGF trigger different biological responses than high concentrations. Therefore, it would be worth to analyze the luminal-like versus stromal interactions using CellPhoneDB and see whether some specific ligand receptor pair would pop up in the analysis.

This is an interesting idea; following the Referee's advice, we have also applied CellPhoneDB to analyze the ligand-receptor pairs between luminal-like cells at E15.5 and both populations of mesenchymal cells (included in revised Figure 4). In our previous version of the manuscript, we had already highlighted the TGF β 2-TGF β R1 and TGF β 2-TGF β R2 interaction pairs between basal-like MECs and dermal mesenchymal cells. The new CellPhoneDB predictions we have implemented found that these interaction pairs are also significant between luminal-like MECs and dermal mesenchymal cells. Of interest, the TGF β 3-TGF β R3 interaction pair was instead only significant between basal-like MECs and dermal mesenchymal cells, whereas it was not predicted with the luminal-like MECs. Likewise, Pthlh signaling is only specific to the communication between basal-like cells and stroma as it is not expressed in luminal-like cells. Other examples include differences in NRP1-SEMA3A, PDGFb-PDGFRa/B and RSPO3/4-LGR4/6 bi-directional interactions between the distinct stromal and epithelial compartments. However, no highly significant interaction between luminal-like cells and stroma could be detected in this new analysis, suggesting that the cells located in the basal position are the main drivers of crosstalk with the stroma. Further conclusions drawn from this analysis are described in the next point (point # 3 from Referee # 3), since we have also compared the interaction pairs at E15.5 with the analysis obtained at E13.5.

3) How is the ligand/receptor interaction landscape changing between basal cells and the mesenchyme at E14.5, before the switch occurs as compared to E15.5 during the potency switch? It is worth comparing the two stages as it would narrow down the factors involved in the epithelial / stromal crosstalk. Also, I do not know to what extent the limited number of cells profiled by scRNA-Seq limits the power of this study, and also how other tools such as NicheNet (<https://www.nature.com/articles/s41592-019-0667-5>) would perform with such dataset.

We thank the Referee for these exciting suggestions. To address how the interaction landscape changes before and during the potency switch, we now provide CellPhoneDB analysis at E13.5 and E15.5 (in revised Figure 4) and compare the results between these time points at lines 340-361. We chose to use the E13.5 instead of E14.5 dataset, because we were concerned that the fewer cells sequenced at E14.5 might give different outputs due to the differences in number of cells sequenced at E14.5 and E15.5, and not because of biological differences.

Several components of IGF signaling were implicated as important mediators of communication between dermal mesenchyme and MECs at both E13.5 and E15.5. Of interest, the IGF1-IGF1R pair is highly significant between both mesenchyme and MECs at E15.5, but not at E13.5, while the reverse is observed for the pairs involving IGF2, only significant at E13.5.

Regarding WNT signaling, many Wnt pathway components were highlighted as being involved at both developmental timepoints in a bidirectional manner. Interestingly, we observed some specific ligand and receptor expression patterns. For instance, Fzd4 (restricted to dermal mesenchyme) and Wnt5a are expressed in mesenchymal clusters, while Wnt3, 4, 7b and 10b signals come from the epithelial clusters. As mentioned above, Wnt3 is expressed by BC-like and not LC-like cells (Figure 4B). Therefore, the specific interaction between Wnt3 and its receptor Fzd1 was highly significant between basal-like MECs and both populations of mesenchymal cells at both developmental timepoints, E13.5 and E15.5.

When analyzing the ligands and receptors linked to TGF β signaling, we found differences between E13.5 and E15.5. Intriguingly, the specific interaction between the TGF β 1 ligand and its receptor TGF β R3 is highly significant between MECs and the sub-epithelial mesenchyme at E13.5, while at E15.5 this interaction is predicted only with the dermal mesenchyme. Likewise, the TGF β 2-TGF β R2 interaction is only detected in the E15.5 epithelium. Of interest, the TGF β 3-TGF β R3 interaction pair is only significant between BC-like cells and dermal mesenchyme, whereas it is not predicted in LC-like cells.

Furthermore, Neuropilins (NRP) interact with VEGF and semaphorins (SEMA) and play a role in angiogenesis, axon guidance, cell survival, migration and invasion. At E13.5, a NRP1-SEMA3a interaction is predicted between MECs and the sub-epithelial mesenchyme, whereas at E15.5, it is specifically found between BC-like (and not LC-like) cells and both mesenchyme compartments. The NRP2-SEMA3f interaction pair is found at both developmental timepoints although it appears attenuated at E15.5.

In this study, we investigated the specific interaction between FGF10 and its receptor FGFR2, which was highly significant between MECs and dermal mesenchymal cells. This crosstalk was also found between FGF18 and FGF2 and the FGFR2 receptor. However, while FGF10-FGFR2 and FGF18-FGFR2 interaction pairs were predicted at both E13.5 and E15.5 timepoints, FGF2-FGFR2 was only found at E15.5. In addition, FGF10, FGF18 and FGF2 ligands are exclusively expressed by dermal mesenchymal cells, whereas FGF7 is expressed in both sub-epithelial and dermal mesenchyme at E13.5 and E15.5.

Finally, as briefly mentioned above, we analyzed the PTHrP signaling pathway, known to govern the transition from a mammary bud to a branched ductal structure. PTHLH, expressed in the epithelium, signals to mesenchymal PTH1R to modulate Wnt signaling (Macias and Hinck, 2012). Specifically, PTHLH is expressed by basal-like cells at E15.5 and acts on its receptor PTH1R, expressed in both the sub-epithelial and dermal mesenchyme (Figure 4A). The interaction PTHLH-PTH1R can already be observed at E13.5. PTHLH can modulate Wnt and BMP signaling in early mammary development and induce the production

of a specialized condensed mesenchyme that maintains mammary epithelial cell fate (Macias and Hinck, 2012). Indeed, the bone morphogenetic protein 4 (BMP4), that we find specifically enriched in the sub-epithelial mesenchyme at E13.5, signals through BMPRI1A (mostly in BC-like cells) to inhibit hair follicle formation in the developing nipple sheath (review E. Spina and P. Cowin, 2021).

Altogether, the comparative analysis between E13.5 and E15.5 recommended by this Referee has underscored the importance of several developmental signaling pathway components, including IGF, WNT, TGF β , NRP and FGF, in the communication between MECs and the surrounding mesenchyme, with specific differences before and after the multipotency switch (from E13 to E15). We have now included the results of this analysis in revised Figure 4 and in the revised text (lines 340-361).

Finally, according to the Referee's request, we also tested the tool NicheNet on our datasets. The main difference between the CellPhoneDB and NicheNet algorithms is that the latter considers the expression of ligands and receptors alongside the target genes of the pathway supposedly activated by a given ligand. When we ran NicheNet on our scRNAseq dataset, however, we found limited overlap with the output obtained with the CellPhoneDB algorithm. When we investigated the reasons for this discrepancy, we realized that, although some pathways were clearly identified by both algorithms, NicheNet failed to rank highly three signaling pathways known to be particularly functionally important for mammary embryonic development (such as the Pthlh, FGF and Wnt pathways) (Figure R6). Based on the lack of robust output, we would prefer not to include the NicheNet analysis in our manuscript.

Figure for reviewers removed

4) Additionally, the authors could go back to their scRNA-Seq dataset, extract ligand encoding genes expressed at high level in both sub epithelial and dermal mesenchyme compartments, and functionally annotate them. This would reinforce the CellphoneDB analysis and help narrowing down the stromal factors that actively contribute to fate decision and branching. For example, is FGF10 expressed at all in the mesenchyme (check the scRNA-Seq dataset)? And is FGFR2 expressed in the basal compartment? In the luminal compartment? Also, multiple Wnt/Fzd pairs also pop up in the CellphoneDB analysis, but what Wnt ligands (if any) are actually expressed by the mesenchyme?

We agree that this approach can reinforce the CellPhoneDB analysis; accordingly, we have functionally annotated by GO analysis ligand encoding genes highly expressed in the stromal clusters (Figure R7). GO analysis revealed that ligand encoding genes that participate in the interaction between sub-epithelial (Figure R7A) or dermal (Figure R7B) mesenchyme and BC-like cells are, as expected, implicated in gland morphogenesis and development; however, GO terms related to cell migration and regulation of epithelial cell proliferation were enriched in the dermal mesenchyme, possibly due to the elevated expression of mitogenic proteins such as Pleiotropin (PTN, see revised Figure 4A). On the other hand, the GO analysis suggested by the Referee supports the idea that short-range signals coming from the sub-epithelial mesenchyme contribute to epithelial branching morphogenesis. Related to the point about cell fate specification, we have not found a significant fold enrichment in GO terms related to basal or luminal differentiation in the stroma, suggesting that commitment to the two lineages is mostly epithelial cell-autonomous, as also implied by our new results with an FGFR inhibitor, that abrogates branching without affecting lineage segregation (see revised Figure 5H-I).

Figure for reviewers removed

Regarding FGF10 expression, we have assessed its levels in our UMAP clusters and found that it is restricted to the dermal mesenchyme (Figure R8A and B, cluster in grey), while *Fgfr2* presents a relatively uniform expression in the 3 mammary epithelial clusters at E15.5 (Figure R8C and D), indicating that all MECs are competent to respond to FGF signals. We have included these plots in revised Figure 4C. Likewise, we now provide the UMAP representation of the expression of the Wnt pathway components identified by CellPhoneDB analysis within the different mesenchymal or epithelial clusters in Figure 4B.

Figure for reviewers removed

5) The authors investigate how FGF10/FGFR2 crosstalk drives branching using a **very powerful ex vivo model of explant culture**. However, the contribution of FGF10 to branching in the mammary gland already received several experimental support, such as : <https://www.ncbi.nlm.nih.gov/pmc/articles/PMC4199126/>, <https://pubmed.ncbi.nlm.nih.gov/9428423/>, <https://www.frontiersin.org/articles/10.3389/fcell.2020.00415/full>, <https://journals.biologists.com/dev/article/129/1/53/17506/Role-of-FGF10-FGFR2b-signaling-during-mammary>

We thank the Referee for their appreciation of our *ex vivo* model of embryonic explant cultures. We agree that the contribution of FGF10 to mammary and lung branching morphogenesis was previously reported, mainly from the insightful work of the Bellusci lab. Indeed, we had cited previous work on embryonic mammary development in our previous version (now at lines 361-364):

“To functionally assess the validity of this computational predictions, we sought to investigate the impact of exogenous FGF10 on embryonic branching morphogenesis, given its reported role in mammary placode development in constitutive FGF10 knock-out mice (Mailleux et al. 2002; Veltmaat et al. 2006).”

We have now included a recent study reporting the combined role of bFGF and FGF10 in promoting branch elongation in salivary gland organoids (Kim et al. 2021) at line 365.

FGF10/FGFR2 signalling is known to be a central player in postnatal mammary branching morphogenesis. Nonetheless, the importance of this signalling axis on dynamic embryonic mammary branching remained unexplored, limited by the fact that only constitutive KO mice for *Fgfr2b* or *Fgf10* existed, and they fail to develop mammary placodes, suggesting a requirement to **initiate** embryonic mammary development. Thus, to overcome this hurdle and **dynamically** investigate the effect of FGF10 or FGFR inhibition on embryonic mammary morphogenesis for the first time, we turned to live cell imaging of explant cultures of mammary embryonic buds that can extensively branch *ex vivo*. Only by combining this approach with time-lapse microscopy and our custom image analysis pipeline, we could discover that FGF10 accelerates embryonic mammary branching independently of any effect on cell proliferation or explant growth.

This is limiting the novelty of the study in this regard, and the validation of the whole scRNA-Seq profiling. Also, in their model, FGF10 ligands from the stroma of the explant might also be driving branching and can mask the biological effects of exogenously added FGF10. Hence, wouldn't it be more suitable to use a FGFR inhibitor (at a sub lethal dose) (or FGF10 blocking ab?) and assess how branching is affected *ex vivo*? How cell fate is affected (with for example absence of lineage switch and/or presence of cell bearing markers of both lineage)? Indeed, the fact that FGF10 signalling contributes to branching does not necessarily mean it drives the switch from bipotency to unipotency.

This is a great suggestion. We have now tested a potent pan-FGFR inhibitor (BGJ398) on embryonic explants and found that exposure to this inhibitor either completely abolishes, or severely impairs, the capacity of mammary buds to branch. It is noteworthy that this is not accompanied by compromised cell fate specification and lineage segregation, suggesting that cell fate commitment does not require branching. These results are now included in the revised manuscript in Fig. 5H-I and in the text at lines 400-414 (Results) and 497-501 (Discussion).

The reasons why the potency switch and the branching coincide remain yet to be elucidated. Adding FGF10 *ex vivo* exposes the luminal like and basal like cells to the same concentration of FGF10, meaning their position in regards to the stroma (and therefore the concentration of FGF10 to which they are exposed to *in vivo*) does not matter anymore, can the authors comment on that ? Finally, given that the number of branches in explant does not change upon FGF10 stimulation (only their diameter), the title of figure 5 appears to be inaccurate.

This is a pertinent point; adding FGF10 exogenously in the explant culture medium does expose all epithelial cells to the same ligand concentration; we now comment on this at lines 400-405. On the other hand, FGFR2 is equally expressed in all 3 epithelial clusters at E15.5 (Figure 4C), indicating that all MECs are competent to respond to FGF stimulation. It is also important to note that by time-lapse and single cell tracking we observe extensive cell movements and rearrangements as the explants grow and branches elongate, therefore we do not believe that *in vivo* BC-like and LC-like cells would stay put in their position throughout tissue morphogenesis. Regarding the title of Figure 5, we originally stated that “FGF10 **accelerates** embryonic mammary branching without affecting cell proliferation.” This reflects the fact that FGF10 accelerates branching as quantified by speed of growth and it does not imply an increase in branch formation. However, since the inclusion of the experiments with FGFR inhibition, the title of Figure 5 has been changed to “FGF signaling is required for embryonic mammary branching”.

The title of this work emphasizes that "positional cues underlie cell fate specification", and given the results, it seems it's the position in regard to the mesenchyme that dictates lineage segregation. Therefore, the authors should strengthen this part of the work.

While we believe that our study puts solid premises for addressing how positional cues within the epithelial buds and factors from the surrounding environment can direct mammary epithelial cell fate specification, we appreciate the Referee's point and have modified our title accordingly to: “Single cell analysis reveals spatially distinct epithelial and mesenchymal cell subsets in the branching embryonic mammary gland”.

Minor concerns:

1) Regarding the gating strategy (Fig Sup 1), how can the authors be sure that all the embryonic mammary

epithelial cells are strictly Epcam high? What about Epcam mid, or low as found in normal adult mammary gland? Can the author comment on the gating strategy?

We apologize for not being clearer on the gating strategy we used to enrich for embryonic MECs. The gates shown in Figure S1A were the ones we used at P0, but we have now included also the gating strategy used at E15.5 (Figure R9 – Expanded view Figure EV1A), where it can be appreciated that we did not select only EpCam high cells, but rather included all Epcam positive cells (high, medium and low) as mammary epithelial cells (MECs). We would like to note that the expression levels of Epcam at early embryonic stages (E13, E14 and E15) are not changed within different epithelial cells, as the segregation of this marker happens only later in development.

Figure for reviewers removed

2) The experimental data from the literature supporting the bi to unipotent progenitor switch post E15.5, including this article, is very convincing. However, I recommend that the discussion of this article is tempered by at least two key points : 1) the existence of bipotent adult MaSC have received several experimental support from different labs and have been associated with different markers such as Dll1 (<https://www.science.org/doi/10.1126/science.aan4153>), PROCN (<https://pubmed.ncbi.nlm.nih.gov/25327250/>), or Krt5 (<https://www.nature.com/articles/nature12948>). The controversy regarding their existence and activity in the adult MG therefore remains a topic of vivid discussion, and it would be adequate to at least place this work in the broader context. 2) how do these findings relate to the human breast biology where MaSC are described in the adult gland, and where cells co-expressing luminal and basal markers have been clearly defined (notably by the work of Tan Ince and others)? How is this bi to unipotency switch relevant to the human biology?

Regarding the potential existence of bipotent MaSCs in the adult gland, the 3 studies cited above did not use lineage-exclusive promoters for fate mapping. This includes the Krt5-CreERT2 model used in Rios et al, which non-specifically labels approx. 1% of the luminal lineage (unlike the Krt5-CreERT2 line generated by the Blanpain's lab in 2011). Thus, the lineage tracing results obtained in these studies are unfortunately not definitive, and the body of work published since the original study by Van Keymuelen et al in 2011 provide strong evidence that adult bipotent MaSCs (if they indeed exist) do not contribute to postnatal mammary gland development and homeostasis. The field has therefore largely achieved a consensus on the potency and activity of adult mammary basal and luminal progenitors post birth. As a result, we prefer to avoid framing our study, which is solely focused on embryonic mammary cell fate dynamics, in this context in the manuscript discussion, and instead emphasise how the identification of new specific promoters of early mammary progenitors in our work will guide future lineage tracing studies to definitively establish the differentiation dynamic and potency of early mammary progenitors and their contribution to postnatal mammary gland development (lines 465-468).

Regarding the second point related to human breast, we now discuss this aspect in the revised text by referring to the existence of cells co-expressing luminal and basal adult markers in the human tissue, as well

as citing the insightful work of Tan Ince (Dontu and Ince, 2015) (lines 468-475). However, it should be noted that recent scRNAseq studies in human breast have not been able to identify a hybrid/mixed cluster, indicating that cells co-expressing some lineage markers (i.e. K8/K14) still hold the identity of BCs or LCs, thus do not form a new UMAP cluster (Nguyen et al, 2018; DOI: 10.1038/s41467-018-04334-1). Moreover, in the absence of lineage tracing experiments, it is impossible to conclude on the potency of these rare cells found in human breast tissues. We believe that elucidating the mechanisms underlying the **switch from multipotency to unipotency** is very important as it is a feature observed in stem cells and progenitors of many other tissues, as well as being very relevant to the human biology because it underlies the mechanisms used to restrict cellular plasticity, which lies at the origin of cancer.

3) Figure 2 as it is, provides limited information and could be put in supplementary figures, while Figure S4 deserved to be among the main figures, together with more in depth analysis of the ligand/receptor interactions. Figure 6 could also be merged with Figure 5.

We thank the Referee for these suggestions. Accordingly, we have performed the proposed changes in the position of the Figures: the previous Figure 2 is now Expanded view Figure EV2 and the previous Figure S4 is now revised Figure 4. However, given we have expanded Figure 5 in the revised manuscript to include our new experiments with the FGFR inhibitor, we prefer to keep Figure 5 and 6 separated.

Dear Silvia, dear Bethan,

Thank you for sharing your revised manuscript (EMBOJ-2023-115114R-Q) for consideration by The EMBO Journal, as well as for your patience with our response at this time of the year. Your amended study was sent back to the referees for their re-evaluation, and we have received comments from two of them, which I enclose below.

As you will see, expert #1 remains critical regarding requirement of lineage tracing and molecular depth presented. In contrast, reviewer #3 acknowledges the revision work adding important reanalyses of the single-cell data and functional work on the FGF inhibition, and is now largely supportive of publication. Please note that while referee #2 was at this time not able to reassess your amended work we found your response to this expert to be reasonable and the critique to be sensibly addressed.

We have discussed all input at hand carefully in the editorial team and now concluded that in light of the strong endorsement by referees #2 and #3 we can offer to proceed with this work towards acceptance.

While expert #1's points are well taken as such, we think further work into these directions is not required for publication of this study.

Thus, we are pleased to inform you that your manuscript has been accepted in principle for publication in The EMBO Journal.

We now need you to take care of a number of issues related to formatting and data presentation, which should be addressed at re-submission, as detailed below.

Please contact me at any time if you have additional questions related to below points.

As you might have seen on our web page, every paper at the EMBO Journal now includes a 'Synopsis', displayed on the html and freely accessible to all readers. The synopsis includes a 'model' figure as well as 2-5 one-short-sentence bullet points that summarize the article. I would appreciate if you could provide this figure and the bullet points.

Thank you for giving us the chance to consider your manuscript for The EMBO Journal. I look forward to your final revision.

Again, please contact me at any time if you need any help or have further questions.

Best regards,

Daniel Klimmeck

>> Please add up to five keywords to your study.

>> Author Contributions: Please remove the author contributions information from the manuscript text. Note that CRediT has replaced the traditional author contributions section as of now because it offers a systematic machine-readable author contributions format that allows for more effective research assessment. and use the free text boxes beneath each contributing author's name to add specific details on the author's contribution.

More information is available in our guide to authors.
<https://www.embopress.org/page/journal/14602075/authorguide>

>> Adjust the title of the 'Conflict of Interest' section to 'Disclosure and Competing Interests Statement' and move after Acknowledgements.

>> Please provide source data for the study as to the separate request e-mail by my colleague Hannah Sonntag.

>> Recheck publication status of preprints Lang, Qiang et al (2023) and Fankhaenel et al (2021) and adjust references in case.

>> Figure callouts: Table 2 should be called out after Table 1; panels A and B of Fig 6 need to be called out.

>> EV figures: there are currently six EV figures. Please limit to five. If required include the additional information into an appendix .pdf file with a ToC on its first page. Nomenclature of figures and tables adjusted to "Appendix Figure S1, S2..." "Appendix Table S1" etc. Please correct textual callouts accordingly.

>> Please remove the heading "Reagents and Tools tables"

>>Data availability section: provide a URL for the GSE dataset.

>> Consider additional changes and comments from our production team as indicated below:

- Data Availability section:

Please note that the specific URL for GSE210594 dataset is not provided in the data availability statement.

- Figure legends:

1. Please note that a separate 'Data Information' section is required in the legends of figures 2d, g, i; EV 4c, e-h; EV 6a-f.
2. Please note that a separate 'Data Information' section is required in the legends of figures 2d, g, i; EV 4c, e-h; EV 6a-f.
3. Please define the annotated p values ****/** in the legend of figure 2f, k; 5b; as appropriate.
4. Please indicate the statistical test used for data analysis in the legends of figures 2f, k; 4a; 5b; EV 1g; EV 3.
5. Please note that in figures 5e-g; there is a mismatch between the annotated p values in the figure legend and the annotated p values in the figure file that should be corrected.
6. Please note that the box plots need to be defined in terms of minima, maxima, centre, bounds of box and whiskers, and percentile in the legend of figure EV 1g.
7. Please note that information related to n is missing in the legends of figures 1f; 2f, k-l; 3b; 4c; 5c, e-g, i; EV 1d, g; EV 5a.
8. Although 'n' is provided, please describe the nature of entity for 'n' in the legend of figure 5b.
9. Please note that the error bars are not defined in the legends of figures 2f, k-l; 5b-c, e-g, i.
10. Please note that the scale bar needs to be defined for figures 5d, h.

Referee #1:

Thank the authors for taking efforts to address my concerns. However, I still think that the revised manuscript does not provide sufficient insights into the molecular mechanisms by which the lineage-restricted cell fate is established at E15.5. Moreover, I believe that lineage tracing studies are necessary to support the primary conclusion in this manuscript. I am glad to hear that the authors are developing two CreERT2 mouse lines to address this. Hope they would be able to provide evidence from these models to conclude that basal- and luminal-restricted bipotent cells are indeed dedicated at as early as E15.5.

Referee #3:

Peer review Carabana et al., round 2.

First of all, I would like to acknowledge the substantial amount of work performed by the authors in this revised manuscript and the care put into addressing thoroughly each point I've raised in my initial review. The overall study remains rather descriptive but provides a definitive timing as to when lineage segregation occurs. It also lays solid ground for investigating functionally additional ligand/receptor crosstalk involved in lineage segregation and/or branching in the mammary gland. In this regard, the new data they provide using the FGFR inhibitor (Figure 5) is very interesting, as it appears that branching and lineage segregation can be decoupled. Indeed, while branching is dramatically impaired in presence of FGFR inhibitor, lineage segregation can still occur, as shown by P63 and K8 staining. Besides, with this explant culture model, the authors have a powerful tool to inhibit or activate additional signaling pathways found in CellPhone analyses, and address their contribution to branching and differentiation.

The authors also emphasize the importance of the crosstalk between the epithelial cells and the neighboring mesenchyme, which is according to me the real strength and novelty of this study. In the revised manuscript, the two types of mesenchyme

(sub-epithelial and dermal) have been better characterized in terms of markers, elegantly validated by immunofluorescence (Figure 3C-E). Additionally, the authors present more in-depth characterization of the ligand/receptor interactome between mesenchyme and the epithelial subsets at different time points, while in the initial version the analysis was restricted to E15.5 with the basal cells, once the segregation had occurred actually. Indeed, the authors add an analysis at E13.5 between the EMPs and the two types of mesenchyme, which provides clues as to what could be the initiating signals (For example IGF, which pops up only at E13.5 and not later). They also add a ligand/receptor analysis between the luminal cells and the mesenchyme (at E15.5), which did not show much qualitative differences, as compared to the analysis with the basal cells. This is suggesting that it might just be a concentration effect; the cells in the middle might be exposed to lower amount of input signals as compared to the cells in the outer layer, which therefore may orient their basal differentiation, which in turn can crosstalk with the luminal cells to initiate their differentiation. Indeed, they also provide a ligand/receptor analysis between luminal and basal cells and revealed that crosstalk between epithelial cells (not only with the mesenchyme) may also contribute to fully accomplish lineage segregation, especially at later developmental stage (Figure EV3, DLL1/Notch 2 and 3 seems to be relevant at P0 and not at E15.5, while PTN/PTPRS seems to be relevant only at E15., for example). Although, these algorithms like CellPhoneDB have their limitations and biases, and although lots of functional validation remains to be performed (beyond the scope of this study), the new manuscript provides an in-depth exploration of their scRNA-Seq dataset giving a more comprehensive and dynamic view (across time, and across the different spatial compartments) of the interactions between the different epithelial and mesenchymal subsets, that may be responsible for branching and lineage segregation. Finally, the authors discuss these novel analyses and findings using the relevant literature which give more substance to the manuscript.

In conclusion, the authors have answered all my comments thoroughly, and their claims are supported by robust data. Although the FGF10 validation for branching is not fully novel owing to the existing literature, most of the findings represent a valuable addition to the field:

- 1) the definitive timing of lineage segregation in the mammary gland, together with a spatial localization of the different cell populations, including the mesenchyme.
- 2) the (improved) characterization and contribution of the mesenchyme at embryonic stage for branching initiation
- 3) the in-depth re-analysis of their scRNAseq which present an exhaustive and comprehensive overview of the crosstalks potentially at stake during embryogenesis, inducing branching and lineage commitment.

According to me, this manuscript is now suitable for a publication at EMBO J.

The authors addressed the editorial issues.

Dear Silvia, dear Bethan,

Thank you for submitting the revised version of your manuscript. I have now evaluated your amended manuscript and concluded that the remaining minor concerns have been sufficiently addressed.

I am pleased to inform you that your manuscript has been accepted for publication in the EMBO Journal.

On a different note, I would like to alert you that EMBO Press offers a format for a video-synopsis of work published with us, which essentially is a short, author-generated film explaining the core findings in hand drawings, and, as we believe, can be very useful to increase visibility of the work. Please see the following link for representative examples and their integration into the article web page:

<https://www.embopress.org/doi/full/10.15252/emboj.2019103932>

Finally, we have noted that the submitted version of your article is also posted on the preprint platform bioRxiv. We would appreciate if you could alert bioRxiv on the acceptance of this manuscript at The EMBO Journal in order to allow for an update of the entry status. Thank you in advance!

Kind regards,

Daniel

Daniel Klimmeck, PhD
Senior Editor
The EMBO Journal
EMBO
Postfach 1022-40
Meyerhofstrasse 1
D-69117 Heidelberg
contact@embojournal.org
Submit at: <http://emboj.msubmit.net>
